# Object-Centric Learning for Real-World Videos by Predicting Temporal Feature Similarities

**Andrii Zadaianchuk**[1,2*]  **Maximilian Seitzer**[1*]  **Georg Martius**[1]

[1] Max Planck Institute for Intelligent Systems, Tübingen, Germany
[2] Department of Computer Science, ETH Zurich
`andrii.zadaianchuk@tuebingen.mpg.de`

## Abstract

Unsupervised video-based object-centric learning is a promising avenue to learn structured representations from large, unlabeled video collections, but previous approaches have only managed to scale to real-world datasets in restricted domains. Recently, it was shown that the reconstruction of pre-trained self-supervised features leads to object-centric representations on unconstrained real-world image datasets. Building on this approach, we propose a novel way to use such pre-trained features in the form of a temporal feature similarity loss. This loss encodes semantic and temporal correlations between image patches and is a natural way to introduce a motion bias for object discovery. We demonstrate that this loss leads to state-of-the-art performance on the challenging synthetic MOVi datasets. When used in combination with the feature reconstruction loss, our model is the first object-centric video model that scales to unconstrained video datasets such as YouTube-VIS. `https://martius-lab.github.io/videosaur/`

## 1 Introduction

Autonomous systems should have the ability to understand the natural world in terms of independent entities. Towards this goal, unsupervised object-centric learning methods [1–3] learn to structure scenes into object representations solely from raw perceptual data. By leveraging large-scale datasets, these methods have the potential to obtain a robust object-based understanding of the natural world. Of particular interest in recent years have been video-based methods [4–7], not least because the temporal information in video presents a useful bias for object discovery [8]. However, these approaches are so far restricted to data of limited complexity, successfully discovering objects from natural videos only on closed-world datasets in restricted domains.

In this paper, we present the method ***Video Slot Attention Using temporal feature similaRity***, VideoSAUR, that scales video object-centric learning to unconstrained real-world datasets covering diverse domains. To achieve this, we build upon recent advances in image-based object-centric learning. In particular, Seitzer et al. [9] showed that reconstructing pre-trained features obtained from self-supervised methods like DINO [10] or MAE [11] leads to state-of-the-art object discovery on complex real-world images. We demonstrate that combining this feature reconstruction objective with a video object-centric model [5] also leads to promising results on real-world YouTube videos.

We then identify a weakness in the training objective of current unsupervised video object-centric architectures [4, 7]: the prevalent reconstruction loss does not exploit the temporal correlations existing in video data for object grouping. To address this issue, we propose a novel self-supervised loss based on *feature similarities* that explicitly incorporates temporal information (see Fig. 1). The loss works by predicting distributions over similarities between features of the current and future

---

[*]equal contribution

37th Conference on Neural Information Processing Systems (NeurIPS 2023).

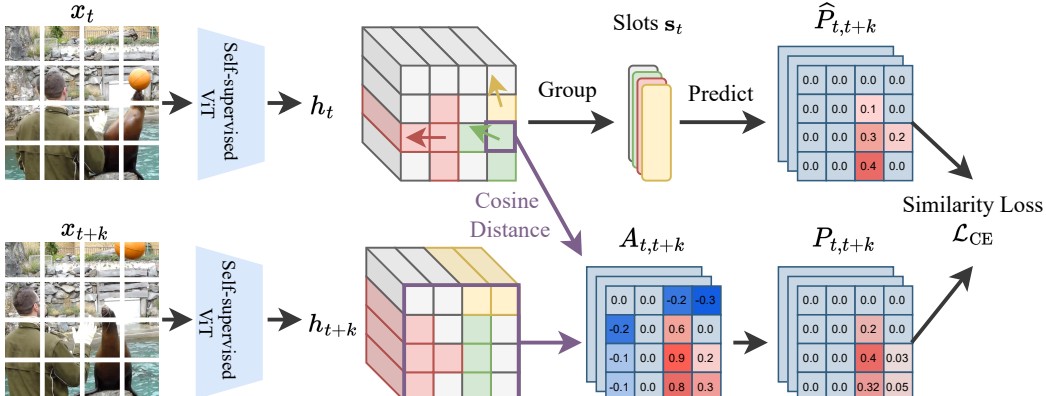

Figure 1: We propose a *self-supervised temporal similarity loss* for training object-centric video models. For each patch at time $t$, the model has to predict a distribution $\hat{P}_{t,t+k}$ indicating where all semantically-similar patches have moved to $k$ steps into the future. The target distribution $P_{t,t+k}$ is computed with a softmax on the affinity matrix $A_{t,t+k}$ containing the cosine distance between all patch features $h_t$, $h_{t+k}$. The loss incentivizes the model to group areas with consistent motion and semantics into slots.

frames. These distributions encode information about the motion of individual image patches. To efficiently predict those motions through the slot bottleneck, the model is incentivized to group patches with similar motion into the same slot, leading to better object groupings as patches belonging to an object tend to move consistently. In our experiments, we find that such a temporal similarity loss leads to state-of-the-art performance on challenging synthetic video datasets [12], and significantly boosts performance on real-world videos when used in conjunction with the feature reconstruction loss.

In video processing, model efficiency is of particular importance. Thus, we design an efficient object-centric video architecture by adapting the SlotMixer decoder [13] recently proposed for 3D object modeling for video decoding. Compared to previous decoder designs [3], the SlotMixer decoder scales gracefully with the number of slots, but has a weaker inductive bias for object grouping. We show that this weaker bias manifests in optimization difficulties in conjunction with conventional reconstruction losses, but trains robustly with our proposed temporal similarity loss. We attribute this to the *self-supervised nature* of the similarity loss: compared to reconstruction, it requires predicting information that is not directly contained in the input; the harder task seems to compensate for the weaker bias of the SlotMixer decoder.

To summarize, our contributions are as follows: (1) we propose a novel self-supervised loss for object-centric learning based on temporal feature similarities, (2) we combine this loss with an efficient video architecture based on the SlotMixer decoder where it synergistically reduces optimization difficulties, (3) we show that our model improves the state-of-the-art on the synthetic MOVi datasets by a large margin, and (4) we demonstrate that our model is able to learn video object-centric representations on the YouTube-VIS dataset [14], while staying fully unsupervised. This paper takes a large step towards unconstrained real-world object-centric learning on videos.

## 2 Related Work

**Video Object-Centric Learning**     There is a rich body of work on discovering objects from video, with two broad categories of approaches: tracking bounding boxes [4, 15–17] or segmentation masks [2, 5–7, 18–25]. Architecturally, most recent image-based models for object-centric learning [3, 9, 26] are based on an auto-encoder framework with a latent slot attention grouping module [3] that extracts a set of slot representations. For processing video data, a common approach [5–7, 21, 24] is then to connect slots recurrently over input frames; the slots from the previous frame act as initialization for extracting the slots of the current frame. We also make use of this basic framework.

**Scaling Object-Centric Learning**     Most recent work has attempted to increase the complexity of datasets where objects can successfully be discovered, such as the synthetic ClevrTex [27] and MOVi

datasets [12]. On natural data, object discovery has so far been limited to restricted domains with a limited variety of objects, such as YouTube-Aquarium and -Cars [7], or autonomous driving datasets like WaymoOpen or KITTI [28]. On more open-ended datasets, previous approaches have struggled [29].

To achieve scaling, some works attempt to *improve the grouping module*, for example by introducing equivariances to slot pose transformations [30], smoothing attention maps [31], formulating grouping as graph cuts [32] or a stick-breaking process [33], or by overcoming optimization difficulties by introducing implicit differentiation [34, 35]. In contrast, we do not change the grouping module, but use the vanilla slot attention cell [3].

Another prominent approach is to introduce *better training signals* than the default choice of image reconstruction. For example, one line of work instead models the image as a distribution of discrete codes conditional on the slots, either autoregressively by a Transformer decoder [7, 26], or via diffusion [36, 37]. While this strategy shows promising results on synthetic data, it so far has failed to scale to unconstrained real-world data [9].

An alternative is to step away from fully-unsupervised representation learning by introducing *weak supervision*. For instance, SAVi [5] predicts optical flow, and SAVi++ [6] additionally predicts depth maps as a signal for object grouping. Other works add an auxiliary loss that regularizes slot attention's masks towards the masks of moving objects [8, 38]. Our model also has a loss that focuses on motion information, but uses an unsupervised formulation. OSRT [13] shows promising results on synthetic 3D datasets, but is restricted by the availability of posed multi-camera imagery. While all those approaches improve on the level of data complexity, it has not been demonstrated that they can scale to unconstrained real-world data.

The most promising avenue so far in terms of scaling to the real-world is to *reconstruct features from modern self-supervised pre-training methods* [10, 11, 39, 40]. Using this approach, DINOSAUR [9] showed that by optimizing in this highly semantic space, it is possible to discover objects on complex real-world image datasets like COCO or PASCAL VOC. In this work, we similarly use such self-supervised features, but for learning on video instead of images. Moreover, we improve upon reconstruction of features by introducing a novel loss based on similarities between features.

**Concurrent Work**    Parallel to this work, two more slot attention-based methods were proposed that learn object-centric representations on real-world videos: SMTC [41] and SOLV [42]. SMTC learns to extracts objects from videos by enforcing semantic and instance consistency over time using a student-teacher approach. SOLV extracts per-frame slots using invariant slot attention [30], applies a temporal consistency module and merges slots using agglomerative clustering; the model is also trained using DINOSAUR-style feature reconstruction, but on masked out intermediate frames.

## 3   Method

In this section, we describe the main new components of VideoSAUR — our proposed object-centric video model — and its training: a pre-trained self-supervised ViT encoder extracting frame features (Sec. 3.1), a temporal similarity loss that adds a motion bias to object discovery (Sec. 3.2), and the SlotMixer decoder to achieve efficient video processing (Sec. 3.3). See Fig. 2 for an overview.

### 3.1   Slot Attention for Videos with Dense Self-Supervised Representations

VideoSAUR is based on the modular video object-centric architecture recently proposed by SAVi [5] and also used by STEVE [7]. Our model has three primary components: (1) a pre-trained self-supervised ViT feature encoder, (2) a recurrent grouping module for temporal slot updates, and (3) the *SlotMixer* decoder (detailed below in Sec. 3.3).

We start by processing video frames $\boldsymbol{x}_t$, with time steps $t \in \{1, \ldots T\}$, into patch features $\boldsymbol{h}_t$:

$$\boldsymbol{h}_t = f_\phi(\boldsymbol{x}_t), \quad \boldsymbol{h}_t \in \mathbb{R}^{L \times D} \tag{1}$$

where $f_\phi$ is a self-supervised Vision Transformer encoder (ViT) [43] with pre-trained parameters $\phi$, and $\boldsymbol{x}_t$ is the input at time step $t$. The ViT encoder processes the image by splitting it to $L$ non-overlapping patches of fixed size (e.g. $16 \times 16$ pixels), adding positional encoding, and transforming them into $L$ feature vectors $\boldsymbol{h}_t$ (see App. C.2 for more details on ViTs). Note that the $i$'th feature retains an association to the $i$'th image patch; the features thus can be spatially arranged. Next,

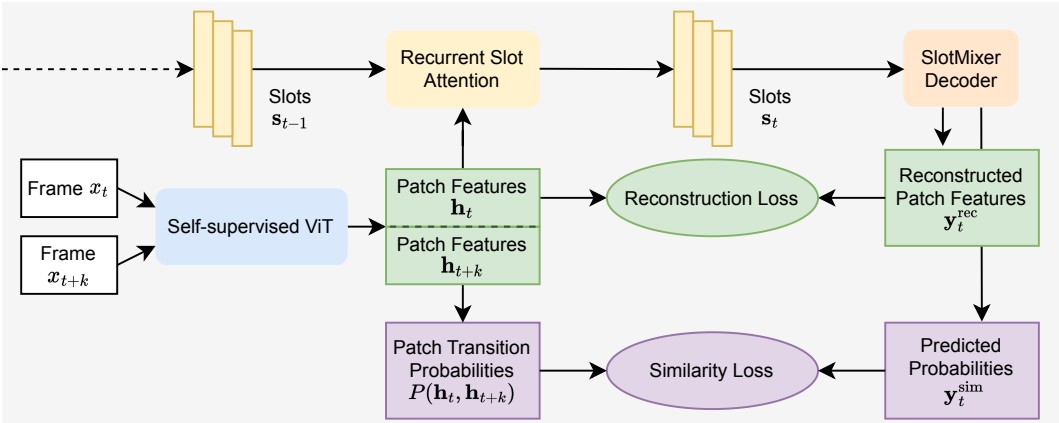

Figure 2: Overview of VideoSAUR. Object slots $s_t$ are extracted from patch features $h_t$ of a self-supervised ViT using time-recurrent slot attention, conditional on slots from the previous time step $t - 1$. The model is trained by reconstructing the patch features $h_t$ of the current frame $x_t$, and by predicting the similarity distribution over patches of a future frame $x_{t+k}$ (see also Fig. 1). The predictions $y_t^{\text{rec}}$ and $y_t^{\text{sim}}$ are decoded efficiently using SlotMixer decoder.

we transform the features from the encoder with a slot attention module [3] to obtain a latent set $s_t = \{s_t^i\}_{i=1}^K$, $s_t^i \in \mathbb{R}^M$ with $K$ slot representations:

$$s_t = \text{SA}_\theta(h_t, s_{t-1}). \tag{2}$$

Slot attention is recurrently initialized with the slots of the previous time step $t - 1$, with initial slots $s_0$ sampled independently from a Gaussian distribution with learned location and scale. Slot attention works by grouping input features into slots by iterating competitive attention steps; we refer to Locatello et al. [3] for more details. To train the model, we use a SlotMixer decoder $g_\psi$ (see Sec. 3.3) to transform the slots $s_t$ to outputs $y_t = g_\psi(s_t)$. Those outputs are used as model predictions for the reconstruction and similarity losses introduced next.

## 3.2 Self-Supervised Object Discovery by Predicting Temporal Similarities

We now motivate our novel loss function based on predicting temporal feature similarities. Video affords the opportunity to discover objects from motion: pixels that consistently move together should be considered as one object, sometimes called the "common fate" principle [44]. However, the widely used reconstruction objective — whether of pixels [5], discrete codes [7] or features [9] — does not exploit this bias, as to reconstruct the input frame, the changes between frames do not have to be taken into account.

Taking inspiration from prior work using optical flow as a prediction target [5], we design a self-supervised objective that requires *predicting patch motion*: for each patch, the model needs to predict where all *semantically-similar* patches have moved to $k$ steps into the future. By comparing self-supervised features describing the patches, we integrate both semantic and motion information; this is in contrast to optical flow prediction, which only relies on motion. Specifically, we construct an affinity matrix $A_{t,t+k}$ with the cosine similarities between all patch features from the present frame $h_t$ and all features from some future frame $h_{t+k}$:

$$A_{t,t+k} = \frac{h_t}{\|h_t\|} \cdot \left(\frac{h_{t+k}}{\|h_{t+k}\|}\right)^\top, \quad A_{t,t+k} \in [-1, 1]^{L \times L}. \tag{3}$$

As self-supervised features are highly semantic, the obtained feature similarities are high for patches that share the same semantic interpretation. Due to the ViT's positional encoding, the similarities also take spatial closeness of patches into account. Figure 3 shows several example affinity matrices.

Because there are ambiguities in our similarity-based derivation of feature movements, we frame the prediction task as *modeling a probability distribution* over target patches — instead of forcing the prediction of an exact target location, like with optical flow prediction. Thus, we define the

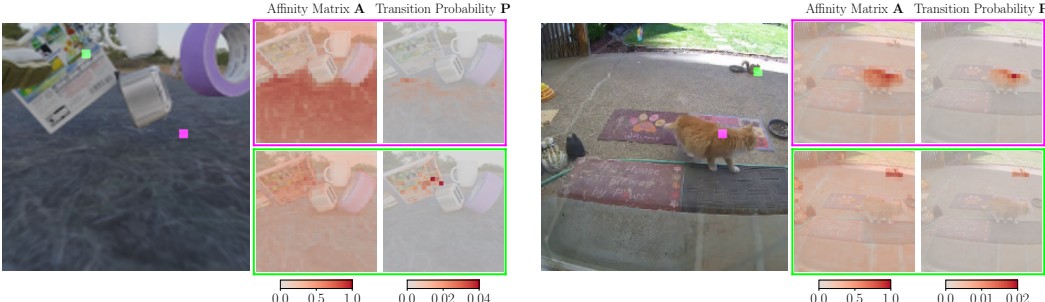

Figure 3: Affinity matrix $\boldsymbol{A}_{t,t+k}$ and transition probabilities $\boldsymbol{P}_{t,t+k}$ values between patches (marked by purple and green) of the frame $\boldsymbol{x}_t$ and patches of the future frame $\boldsymbol{x}_{t+k}$ in MOVi-C (left) and YT-VIS (right). Red indicates maximum affinity/probability. Also see Fig. B.4 for more examples, and our website for an interactive visualization of temporal feature similarities.

probability that patch $i$ moves to patch $j$ by normalizing the rows of the affinity matrix with the softmax, while masking negative similarity values (superscripts refer to the elements of the matrix):

$$\boldsymbol{P}^{ij} = \begin{cases} \dfrac{\exp(\boldsymbol{A}^{ij}/\tau)}{\displaystyle\sum_{k\in\{j|\boldsymbol{A}^{ij}\geq 0\}} \exp(\boldsymbol{A}^{ik}/\tau)} & \text{if } \boldsymbol{A}^{ij} \geq 0, \\ 0 & \text{if } \boldsymbol{A}^{ij} < 0, \end{cases} \tag{4}$$

where $\tau$ is the softmax temperature. The resulting distribution can be interpreted as the *transition probabilities* of a random walk along a graph with image patches as nodes [45]. Then, we define the similarity loss as the cross entropy between decoder outputs and transition probabilities:

$$\mathcal{L}_{\theta,\psi}^{\text{sim}} = \sum_{l=1}^{L} \text{CE}(\boldsymbol{P}_{t,t+k}^l; \boldsymbol{y}_t^l). \tag{5}$$

Figure 1 illustrates the loss computation for an example pair of input frames.

**Why is this Loss Useful for Object Discovery?**    Predicting which parts of the videos move consistently is most efficient with an object decomposition that captures moving objects. This is similar to previous losses predicting optical flow [5]. But in contrast, our loss (Eq. 5) also yields a useful signal for grouping when parts of the frame are *not* moving: as feature similarities capture semantic aspects, the task also requires predicting which patches are semantically similar, helping the grouping into objects e.g. by distinguishing fore- and background (see Fig. 3). Optical flow for grouping also has limits when camera motion is introduced; in our experiments, we find that our loss is more robust in such situations. Methods based on optical flow or motion masks can also struggle with inaccurate flow/motion mask labels — unlike our method, which does not require such labels. This is of particular importance for in-the-wild video, where motion estimation is challenging.

**Role of Hyperparameters.**    The loss has two hyperparameters: the time shift into the future $k$ and the softmax temperature $\tau$. The optimal time shift depends on the expected time scales of movements in the modeled videos and should be chosen accordingly. The temperature $\tau$ controls the concentration of the distribution onto the maximum. Thus, it effectively modulates between two different tasks: accurately estimating the patch motion (low $\tau$), and predicting the similarity of each patch to all other patches (high $\tau$). In particular in scenes with little movement, the latter may be important to maintain a meaningful prediction task. In our experiments, we find that the best performance is obtained with a balance between the two, showing that both modes are important.

**Final Loss.**    While the temporal similarity loss yields state-of-the-art performance on synthetic datasets, as shown below, we found that on real-world data, performance can be further improved by adding the feature reconstruction objective as introduced in Seitzer et al. [9]. We hypothesize this is because the semantic nature of feature reconstruction adds another useful bias for object discovery. Thus, the final loss is given by:

$$\mathcal{L}_{\theta,\psi} = \sum_{t=1}^{T-k} \mathcal{L}_{\theta,\psi}^{\text{sim}}(\boldsymbol{P}_{t,t+k}, \boldsymbol{y}_t^{\text{sim}}) + \alpha\mathcal{L}_{\theta,\psi}^{\text{rec}}(\boldsymbol{h}_t, \boldsymbol{y}_t^{\text{rec}}), \tag{6}$$

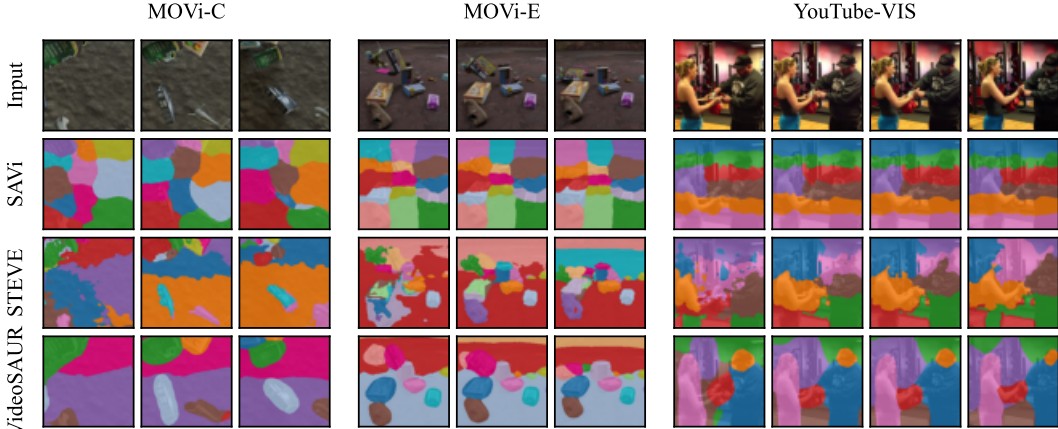

Figure 4: Example predictions of VideoSAUR compared to recent video object-centric methods.

where $\boldsymbol{y}_t = [\boldsymbol{y}_t^{\text{sim}} \in \mathbb{R}^{L \times L}, \boldsymbol{y}_t^{\text{rec}} \in \mathbb{R}^{L \times D}]$ is the output of the SlotMixer decoder $g_\psi$ and $\alpha$ is a weighting factor used to make the scales of the two losses similar (we use a fixed value of $\alpha = 0.1$ for all experiments on real-world datasets). Like in Seitzer et al. [9], we do not train the ViT encoder $f_\phi$.

### 3.3 Efficient Video Object-Centric Learning with the SlotMixer Decoder

In video models, resource efficiency is of particular concern: recurrent frame processing increases the load on compute and memory. The standard mixture-based decoder design [3] decodes each output $K$-times, where $K$ is the number of slots, and thus scales linearly with $K$ both in compute and memory. This can become prohibitive even for a moderate number of slots. The recently introduced SlotMixer decoder [13] for 3D object-centric learning instead has, for all practical purposes, constant overhead in the number of slots, by only decoding once per output. Thus, we propose to use a SlotMixer decoder $g_\psi$ for predicting the probabilities $\boldsymbol{P}_{t,t+k}$ from the slots $\boldsymbol{s}_t$. To adapt the decoder from 3D to 2D outputs, we change the conditioning on 3D query rays to $L$ learned positional embeddings, corresponding to $L$ patch outputs $\boldsymbol{y}_t^l$. See App. C.1 for more details on the SlotMixer module.

As a consequence of the increased efficiency of SlotMixer, there also is increased flexibility of how slots can be combined to form the outputs. Because of this, this decoder has a weaker inductive bias towards object-based groupings compared to the standard mixture-based decoder. With the standard reconstruction loss, we observed that this manifests in training runs in which no object groupings are discovered. But in combination with our temporal similarity loss, these instabilities disappear (see App. B.4). We attribute this to the *self-supervised nature* of the similarity loss[2]; having to predict information that is not directly contained in the input increases the difficulty of the task, reducing the viability of non-object based groupings.

## 4 Experiments

We have conducted a number of experiments to answer the following questions: (1) Can object-centric representations be learned from a large number of diverse real-world videos? (2) How does VideoSAUR perform in comparison to other methods on well-established realistic synthetic datasets? (3) What are the effects of our proposed temporal feature similarity loss and its parameters? (4) Can we transfer the learned object-grouping to unseen datasets? (5) How efficient is the SlotMixer decoder in contrast to the mixture-based decoder?

### 4.1 Experimental Setup

**Datasets** To investigate the characteristics of our proposed method, we utilize three synthetic datasets and three real-world datasets. For synthetic datasets, we selected the MOVi-C, MOVi-D

---

[2]Novel-view synthesis, the original task for which SlotMixer was proposed, is similarly a self-supervised prediction task. This may have contributed to the success of SlotMixer in that setting.

Table 1: Comparison with state-of-the-art methods on the MOVi-C, MOVi-E, and YT-VIS datasets. We report foreground adjusted rand index (FG-ARI) and mean best overlap (mBO) over 5 random seeds. Both metrics are computed for the whole video (24 frames for MOVi, 6 frames for YT-VIS).

| | MOVi-C | | MOVi-E | | YT-VIS | |
|---|---|---|---|---|---|---|
| | FG-ARI | mBO | FG-ARI | mBO | FG-ARI | mBO |
| Block Pattern | 24.2 | 11.1 | 36.0 | 16.5 | 24 | 14.9 |
| SAVi [5] | 22.2 ± 2.1 | 13.6 ± 1.6 | 42.8 ± 0.9 | 16.0 ± 0.3 | 11.1 ± 5.6 | 12.7 ± 2.3 |
| STEVE [7] | 36.1 ± 2.3 | 26.5 ± 1.1 | 50.6 ± 1.7 | 26.6 ± 0.9 | 20.0 ± 1.5 | 20.9 ± 0.5 |
| VideoSAUR | **64.8 ± 1.2** | **38.9 ± 0.6** | **73.9 ± 1.1** | **35.6 ± 0.5** | **39.5 ± 0.6** | **29.1 ± 0.4** |

and MOVi-E datasets [12] that consist of numerous moving objects on complex backgrounds. Additionally, we evaluate the performance of our method on the challenging YouTube Video Instance Segmentation (YT-VIS) 2021 dataset [14] as an unconstrained real-world dataset. Furthermore, we examine how well our model performs when transferred from YT-VIS 2021 to YT-VIS 2019 [46] and DAVIS [47]. Finally, we use the COCO dataset [48] to study our proposed similarity loss function with image-based object-centric learning.

**Metrics** We evaluate our approach in terms of the quality of the discovered slot masks (output by the decoder), using two metrics: video foreground ARI (FG-ARI) [2] and video mean best overlap (mBO) [49]. FG-ARI is a video version of a widely used metric in the object-centric literature that measures the similarity of the discovered objects masks to ground truth masks. This metric mainly measures *how well objects are split*. mBO assesses the correspondence of the predicted and the ground truth masks using the intersection-over-union (IoU) measure. In particular, each ground truth mask is matched to the predicted mask with the highest IoU, and the average IoU is then computed across all assigned pairs. Unlike FG-ARI, mBO also considers background pixels, and provides a measure of *how accurately the masks fit the objects*. Both metrics also consider the consistency of the assigned object masks over the whole video.

In addition, we also use image-based versions of those metrics (*Image FG-ARI* and *Image mBO*, computed on individual frames) for comparing with image-based methods.

**Baselines** We compare our method with two recently proposed methods for unsupervised object-centric learning for videos: SAVi [5] and STEVE [7]. SAVi uses a mixture-based decoder and is trained with image reconstruction. We use the unconditional version of SAVi. STEVE uses a transformer decoder and is trained by reconstructing discrete codes of a dVAE [50]. Similar to Seitzer et al. [9], we also add a regular block pattern baseline, corresponding to splitting the video into regular block masks of similar size that do not change over time. By showing the metric values for a trivial decomposition of the video, this baseline is useful to contextualize the results of the other methods. In addition to video-based methods, we compare our model to image-based methods, including DINOSAUR [9], LSD [36] and Slot Diffusion [37], showing that our approach performs well in both object separation and mask sharpness. Last, we also compare our model to two concurrent works discovering objects from real-world video, SMTC [41] and SOLV [42].

### 4.2 Comparison with State-of-the-Art Object-Centric Learning Methods

When comparing VideoSAUR to STEVE and SAVi, it is evident that VideoSAUR outperforms the baselines by a significant margin, both in terms of FG-ARI and mBO (see Table 1 and Fig. 4). On the most challenging synthetic dataset (MOVi-E), VideoSAUR reaches 73.9 FG-ARI. Notably, for the challenging YT-VIS 2021 dataset, both baselines perform comparable or worse than the block pattern baseline in terms of FG-ARI, showing that previous methods struggle to decompose real-world videos into consistent objects. We additionally compare VideoSAUR to image-based methods in App. A.1, including strong recent methods (LSD, SlotDiffusion and DINOSAUR), and find that our approach also outperforms the prior image-based SoTA. Finally, in App. A.2, we find that our method performs competitively with concurrent work.

Next, we report how well our method performs in terms of zero-shot transfer to other datasets to show that the learned object discovery does generalize to unseen data. In particular, we train VideoSAUR

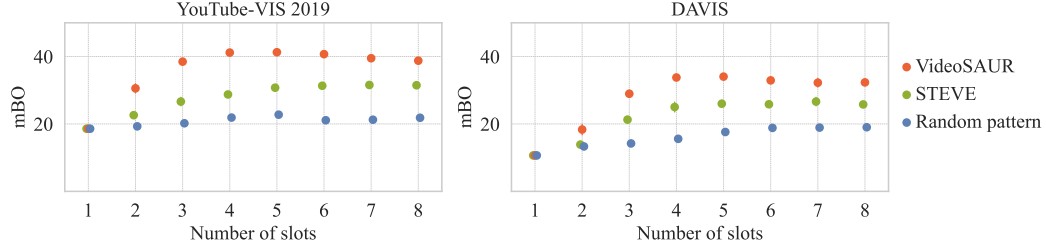

Figure 5: Zero-shot transfer of learned object-centric representations on YT-VIS 2021 to the YT-VIS 2019 and DAVIS datasets for different number of slots.

on the YT-VIS 2021 dataset and evaluate it on the YT-VIS 2019 and DAVIS datasets. YT-VIS 2019 has similar object categories, but a smaller number of objects per image. The DAVIS dataset consists of videos from a fully different distribution than YT-VIS 2021. As the number of slots can be changed during evaluation, we test VideoSAUR with different number of slots, revealing that the optimal number of slots is indeed smaller for these datasets. We find that our method achieves a performance of $41.3 \pm 0.9$ mBO on YT-VIS 2019 dataset and $34.0 \pm 0.4$ mBO on DAVIS dataset (see Fig. 5), illustrating its capability to effectively transfer the learned representations to previously unseen data with different object categories and numbers of objects.

**Long-term Video Consistency** In addition to studying how VideoSAUR performs on relatively short 6-frame video segments from YT-VIS, we also evaluate our method on longer videos. In App. B.1, we show the performance for 12-frame and full YT-VIS videos. While, as can be expected, performance on longer video segments is smaller in terms of FG-ARI, we show that the gap between VideoSAUR and the baselines is large, indicating that VideoSAUR can track the main objects in videos over longer time intervals. Closing the gap between short-term and long-term consistency using memory modules [24, 51] is an interesting future direction that could be useful for video prediction [52] as well as for object-centric goal-based [53, 54] and model-based [55] reinforcement learning.

### 4.3 Analysis

In this section, we analyze various aspects of our approach, including the importance of the similarity loss, the impact of hyperparameters (time-shift $k$ and softmax temperature $\tau$), and the effect of the choice of self-supervised features and decoder.

**Choice of Loss Function (Table 2 and Table 3)** We conduct an ablation study to demonstrate the importance of the proposed temporal similarity loss, comparing and combining it with the feature reconstruction loss [9]. We also consider predicting the features of the *next frame* (see App. C.4 for implementation details). For all datasets, feature reconstruction alone performs significantly worse than the combination of feature reconstruction and temporal similarity loss. Predicting the features of the next frame in addition to feature reconstruction also yields improved performance, but is worse than the temporal similarity, suggesting that the success of our loss can be partially explained by the integration of temporal information through future prediction. Interestingly, on MOVi-C, using the temporal similarity loss alone significantly improves the performance over feature reconstruction ($+20$ FG-ARI, $+7$ mBO). To provide insight into the qualitative differences between the losses, we analyze the videos with the most significant differences in FG-ARI (see Fig. E.4): unlike feature reconstruction, the temporal similarity loss does not fragment the background or large objects into numerous slots, and it exhibits improved object-tracking capabilities even when object size changes. To gain further insights, we also consider (ground truth) *optical flow* as a prediction target that only captures motion, but no semantic information (see App. B.2 for a detailed discussion). We find that only predicting optical flow is not enough for a successful scene decomposition, underscoring the importance of integrating both motion and semantic information for real-world object discovery.

**Robustness to Camera Motion (Table 4)** Next, we investigate if VideoSAUR training with the similarity loss is robust to camera motion, as such motion makes isolating the object motion more difficult. As a controlled experiment, we compare between MOVi-D (without camera motion) and

Table 2: Loss ablation on MOVi-C.

| Loss Type | | | Metric | |
|---|---|---|---|---|
| Feat. Rec. | Next Frame Feat. Pred. | Temp. Sim. | FG-ARI | mBO |
| ✓ | | | 40.2 | 23.5 |
| ✓ | ✓ | | 47.2 | 24.7 |
| | | ✓ | **60.8** | **30.5** |
| ✓ | | ✓ | 60.7 | 30.3 |

Table 3: Loss ablation on YT-VIS.

| Loss Type | | | Metric | |
|---|---|---|---|---|
| Feat. Rec. | Next Frame Feat. Pred. | Temp. Sim. | FG-ARI | mBO |
| ✓ | | | 35.4 | 26.7 |
| ✓ | ✓ | | 37.9 | 27.3 |
| | | ✓ | 26.2 | **29.1** |
| ✓ | | ✓ | **39.5** | **29.1** |

Table 4: Robustness to introducing camera motion (MOVi-D → MOVi-E).

| | MOVi-D | MOVi-E |
|---|---|---|
| SAVi (optical flow) [12] | 19.4 | 2.7 |
| VideoSAUR (temporal sim.) | 55.7 | 62.5 |

Table 5: Decoder comparison on MOVi-C and YT-VIS.

| | MOVi-C | | YT-VIS | | Memory |
|---|---|---|---|---|---|
| | FG-ARI | mBO | FG-ARI | mBO | GB @24 slots |
| Mixer | 60.8 | 30.5 | 39.5 | 29.1 | 24 |
| MLP | 64.2 | 27.2 | 39.0 | 29.1 | 70 |

MOVi-E (with camera motion), and train VideoSAUR using only the temporal similarity loss. We contrast with SAVi trained with optical flow prediction[3], and find that VideoSAUR is more robust to camera motion, performing better on the MOVi-E dataset than on the MOVi-D dataset ($+6.8$ vs $-16.7$ FG-ARI for SAVi).

**Choice of Decoder (Table 5)**    We analyze how our method performs with different decoders and find that both the MLP broadcast decoder [9] and our proposed SlotMixer decoder can be used for optimizing the temporal similarity loss. VideoSAUR with the MLP broadcast decoder achieves similar performance on YT-VIS and MOVi datasets, but requires 2–3 times more GPU memory (see App. C.1 for more details and Table B.3 for the detailed comparison of decoders on MOVI-E dataset). Thus, we suggest to use the SlotMixer decoder for efficient video processing.

**Softmax Temperature (Figure 6a)**    We train VideoSAUR with DINO S/16 features using different softmax temperatures $\tau$. We find that there is a sweet spot in terms of grouping performance at $\tau = 0.075$. Lower and higher temperatures lead to high variance across seeds, potentially because there is not enough training signal with very peaked (low $\tau$) and diffuse (high $\tau$) target distributions.

**Target Time-shift (Figure 6b)**    We train VideoSAUR with DINO S/16 features using different time-shifts $k$ to construct the affinity matrix $\boldsymbol{A}_{t,t+k}$. On both synthetic and real-world datasets, $k = 1$ generally performs best. Interestingly, we find that for $k = 0$, performance drops, indicating that predicting pure self-similarities is not a sufficient task for discovering objects on its own.

**Choices for Self-Supervised Features (Figures 6c and 6d)**    We study two questions about the usage of the ViT features: which ViT features (queries/keys/values/outputs) should be used for the temporal similarity loss? Do different self-supervised representations result in different performance? In Fig. 6c, we observe that using DINO "key" and "query" features leads to significantly larger mBO, while for FG-ARI "query" is worse and the other features are similar. Potentially, this is because keys are used in the ViT's self-attention and thus could be particularly good to compare with the scalar product similarity. Consequently, VideoSAUR uses "key" features in all other experiments. Moreover, we study if the temporal similarity loss is compatible with different self-supervised representations. In Fig. 6d, we show that VideoSAUR works well with 4 different types of representations, with MSN [39] and DINO [10] performing slightly better than MAE [11] and MOCO-v3 [40]. We also demonstrate that *further fine-tuning the DINO features* utilizing a self-supervised temporal-alignment clustering approach named TimeTuning [56] on unlabeled videos enhances the mask quality of VideoSAUR.

**Pre-training Dataset (Table 6)**    All self-supervised methods we utilize are trained on the ImageNet dataset, which a) has a strong bias towards object-centricness as its images mostly contain single objects, and b) introduces a large number of additional images external to the dataset we are training

---

[3]SAVi results with optical flow are from Greff et al. [12].

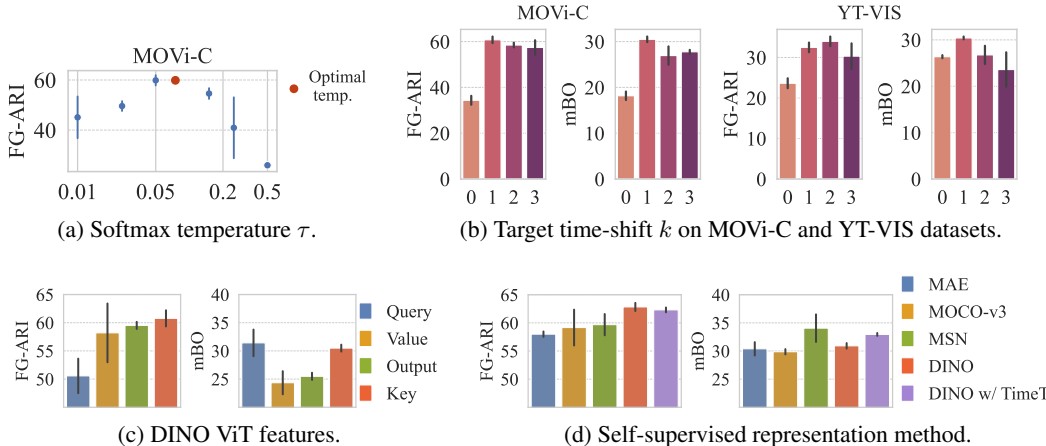

Figure 6: Studying the effect of different parameters of the temporal similarity loss.

Table 6: Comparing VideoSAUR with features trained on MOVi-E (*MAE+MOVi-E*) to features trained on ImageNet (*MAE+ImageNet*). For MAE+MOVi-E, we pre-train a ViT-B/16 using the self-supervised MAE method on MOVi-E for 200 epochs. VideoSAUR is able to perform high-quality object discovery even without access to any external data.

|  | MOVi-C | | MOVi-E | |
|---|---|---|---|---|
|  | FG-ARI | mBO | FG-ARI | mBO |
| VideoSAUR w/ *MAE+ImageNet* features | 58.0 | 30.4 | 72.8 | 27.1 |
| VideoSAUR w/ *MAE+MOVi-E* features | 59.8 | 27.5 | 70.6 | 23.3 |

VideoSAUR on. An interesting question is whether a) and b) are actually required for the success of our method. To answer it, we train a ViT-B/16 encoder from scratch on the MOVi-E dataset using the MAE method, and then train VideoSAUR using the obtained features. Interestingly, we find that the features from MOVi-E yield similar results compared to ImageNet-trained features (although with slight drops in mask quality), demonstrating that VideoSAUR is able to perform high-quality object discovery even without access to external data. This result also has broader implications as it potentially increases the applicability of feature reconstruction-based object-centric methods to datasets fully out of the domain of ImageNet. It also raises a follow-up question: what properties of the pre-training dataset (and method) are important to obtain good target features for object discovery?

## 5   Conclusion

This paper presents the first method for unsupervised video-based object-centric learning that scales to diverse, unconstrained real-world datasets such as YouTube-VIS. By leveraging dense self-supervised features and extracting motion information with temporal similarity loss, we demonstrate superior performance on both synthetic and real-world video datasets. We hope our new loss function can inspire the design of further self-supervised losses for object-centric learning, especially in the video domain where natural self-supervision is available.

Still, our method does not come without limitations: in longer videos with occlusions, slots can get reassigned to different objects or the background (see Fig. B.5 for visualizations of failure cases). VideoSAUR also inherits a limitation of all slot attention-based method, namely that the the number of slots is static and needs to be chosen a priori. Similar to DINOSAUR [9], the quality of the object masks is restricted by the patch-based nature of the decoder. Finally, while the datasets we use in this work are significantly less constrained compared to datasets used by prior work, they still do not capture the full open-world setting that object-centric learning aspires to solve. Overcoming these limitations is a great direction for future work.

## Acknowledgements

We thank Martin Butz, Cansu Sancaktar and Manuel Traub for useful suggestions and discussions. The authors thank the International Max Planck Research School for Intelligent Systems (IMPRS-IS) for supporting Maximilian Seitzer. Andrii Zadaianchuk is supported by the Max Planck ETH Center for Learning Systems. Georg Martius is a member of the Machine Learning Cluster of Excellence, funded by the Deutsche Forschungsgemeinschaft (DFG, German Research Foundation) under Germany's Excellence Strategy – EXC number 2064/1 – Project number 390727645. We acknowledge the support from the German Federal Ministry of Education and Research (BMBF) through the Tübingen AI Center (FKZ: 01IS18039B).

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

# Supplementary Material for Object-Centric Learning for Real-World Videos by Predicting Temporal Feature Similarities

# A  Comparison with Additional Baselines

## A.1  Comparison with Image-Based Object-Centric Methods

In this section, we evaluate how effective our model is for unsupervised *image* segmentation from videos. In addition to reporting results for the video-based object-centric methods SAVi and STEVE, we compare with several recent image-based object-centric learning methods. SLATE [26] is an image-based object-centric model that trains a discrete VAE [50] as a dense feature extractor and uses a Transformer decoder conditioned on slots to reconstruct discrete representations of VAE features. LSD [36] replaces the Transformer decoder with a latent diffusion model conditioned on the object slots. Next, DINOSAUR [9] incorporates dense DINO features as targets and reconstructs the features itself. We report the *Image FG-ARI* and *Image mBO* metrics. They measure how well the predicted segmentation matches the ground-truth segmentation of a given single image (frame), thus consistency over the video is not taken into account.

The results for MOVi datasets are presented in Table A.1. VideoSAUR surpasses both previous image- and video-based methods, showing the benefits of using motion information in combination with semantically coherent self-supervised features. Interestingly, VideoSAUR performs well on both Image FG-ARI and Image mBO metrics, whereas DINOSAUR and LSD seem to improve either in quality of split (measured in Image FG-ARI) or in the sharpness of masks (measured in Image mBO) at the cost of performing worse on the second metric. We also note that LSD results are from larger resolution of MOVi images ($256 \times 256$). We expect that our method can additionally improve if we also use larger resolution videos; however, to be comparable with other baselines, we use $128 \times 128$ resolution in this work.

In addition, we also compare VideoSAUR with DINOSAUR and other video-based baselines on the more challenging YouTube-VIS dataset (see Fig. A.1). VideoSAUR outperforms DINOSAUR ($+4$ FG-ARI) and also surpasses video-based STEVE and SAVi methods. This underscores the benefit of our temporal similarity loss over mere feature reconstruction for challenging real-world datasets.

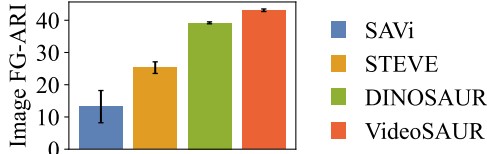

Figure A.1: Image-based comparison on YouTube-VIS (mean ± standard dev., 3 seeds).

Table A.1: Comparison with state-of-the-art methods on the MOVi-C, MOVi-E image datasets. Both metrics are computed for individual frames. The results for SLATE and DINOSAUR are from Seitzer et al. [9], while LSD results are from Jiang et al. [36]. We report mean ± standard dev. over 5 runs for our model.

|  | MOVi-C | | MOVi-E | |
| --- | --- | --- | --- | --- |
|  | Image FG-ARI | Image mBO | Image FG-ARI | Image mBO |
| Block Pattern | 42.7 | 19.5 | 41.9 | 20.4 |
| SAVi [5] | 41.8 | 25.9 | 50.3 | 20.3 |
| STEVE [7] | 51.9 | 41.6 | 59.5 | 34.4 |
| SLATE [26] | 43.6 | 26.5 | 44.4 | 23.6 |
| LSD [36] | 50.5 | 46.3 | 53.4 | 39.6 |
| SlotDiffusion [37] | – | – | 60.0 | 30.2 |
| DINOSAUR [9] | 68.6 | 39.1 | 65.1 | 35.5 |
| VideoSAUR | **75.5 ± 0.9** | **46.0 ± 0.6** | **78.4 ± 0.7** | **41.2 ± 0.4** |

## A.2  Comparison with Concurrent Work on Real-World Videos

In concurrent work, two more slot attention-based methods were proposed that learn object-centric representations on real-world videos: SMTC [41] and SOLV [42]. SMTC learns to extracts objects from videos by enforcing semantic and instance consistency over time using a student-teacher approach. SOLV extracts per-frame slots using invariant slot attention [30], applies a temporal consistency module and merges slots using agglomerative clustering; the model is trained using

Table A.2: Comparison of VideoSAUR with DI-NOv2 ViT B/14 features to SMTC [41] on the DAVIS-2017-Unsupervised validation set. The results for SMTC are from Qian et al. [41].

| Method | $\mathcal{J}$ | $\mathcal{F}$ | $\mathcal{J}$ & $\mathcal{F}$ |
|---|---|---|---|
| SMTC | 36.4 | **44.6** | **40.5** |
| VideoSAUR | **36.8** | 21.1 | 29.0 |

Table A.3: Comparison of VideoSAUR with DINOv2 ViT B/14 features to OCLR [57] and SOLV [42] on YT-VIS 2019. The results for OCLR and SOLV are from Aydemir et al. [42].

| Method | mIoU |
|---|---|
| OCLR | 32.5 |
| SOLV (w/o slot merging) | 39.9 |
| SOLV (w/ slot merging) | **45.3** |
| VideoSAUR | 40.3 |

DINOSAUR-style feature reconstruction on masked out intermediate frames. In this section, we compare to them using the respective evaluation protocols in their papers.

First, we compare to SMTC in Table A.2 on the DAVIS-2017-Unsupervised dataset [47]. Specifically, we assess our method's transfer performance when trained on YT-VIS 201, and follow the evaluation procedure outlined in [47] for matching ground truth masks to predictions using mean $\mathcal{J}$ & $\mathcal{F}$. We report the Jaccard index $\mathcal{J}$ (equivalent to IoU), the boundary F-score $\mathcal{F}$ and their average as $\mathcal{J}$ & $\mathcal{F}$. While the contours of VideoSAUR's segmentation masks (as measured by $\mathcal{F}$) are not as accurate due to processing images and predicting masks at a lower patch resolution ($37 \times 37$ for VideoSAUR trained with DINOv2 B14 features on original DINOv2 resolution), the Jaccard index $\mathcal{J}$ is comparable to SMTC.

Second, we compare to SOLV in Table A.3 on the YT-VIS 2019 dataset. We also list the performance of OCLR [57], which uses synthetic data with ground-truth optical flow. As SOLV, we report the mIoU metric matched over the entire video. The results for VideoSAUR are obtained using random 200 videos from the YT-VIS 2019 train split [4]. Our results surpass OCLR, showing VideoSAUR 's effectiveness in extracting motion information directly from video data using dense self-supervised features. Additionally, our performance matches that of SOLV without using agglomerative clustering, while SOLV with slot merging outperforms VideoSAUR ($+5$ mIoU). This highlights the importance of correctly determining the number of slots. While the main concern in this paper is to integrate motion information from video, we see determining the number of slots as an important orthogonal direction. Thus, combining our method a solution such as the one from Aydemir et al. [42] is an interesting direction for future work.

# B    Additional Experiments

## B.1    Long-Term Video Consistency

Beyond the initial examination of how our VideoSAUR performs on the relatively brief 6-frame video segments from YouTube-VIS 2021, we extend our evaluation to also assess its effectiveness on more substantial, longer video segments. In Table B.1, we show the performance for 12-frame and full YT-VIS video segments (see Fig. B.1 for the distribution of video lengths). Although the performance of VideoSAUR on extended video segments predictably decreases in terms of FG-ARI, the observed difference between VideoSAUR and the baseline models is significant. This suggests that VideoSAUR maintains its efficacy in tracking the primary objects in videos across longer time intervals.

Additionally, we investigate if VideoSAUR benefits from using DINOv2 features [58] that are obtained by training DINO on the larger dataset and fine-tuning the representation on larger resolution ($518 \times 518$). We show that VideoSAUR performance benefits from using such features as a backbone, especially in terms of mask quality ($+6$ mBO points). Using those features with the original resolution VideoSAUR reaching 29.7 mBO on full-length YT-VIS 2021 videos. In addition to this quantitative evaluation, we visualize VideoSAUR predictions on long YT-VIS videos (longer than 30-frames) in Figure E.5 and Figure E.6.

---

[4]We use part of the train split, as validation labels are not released. Exact indexes are available at `validation` split of the Tensorflow version of the YT-VIS 2019 dataset: `https://www.tensorflow.org/datasets/catalog/youtube_vis#youtube_visonly_frames_with_labels_train_split`

Table B.1: Performance of VideoSAUR on the YT-VIS 2021 dataset, varying the length of the video segment (mean ± standard dev., 5 seeds for VideoSAUR with DINO features and STEVE. VideoSAUR with DINOv2 features are one seed only due to computational limitations).

| | 6 frames | | 12 frames | | Full Videos | |
|---|---|---|---|---|---|---|
| | FG-ARI | mBO | FG-ARI | mBO | FG-ARI | mBO |
| Block Pattern | 24.0 | 14.9 | 20.3 | 14.2 | 15.1 | 13.1 |
| STEVE [7] | 20.0 ± 1.5 | 20.9 ± 0.5 | 18.0 ± 1.4 | 21.5 ± 0.5 | 15.0 ± 0.7 | 19.1 ± 0.4 |
| VideoSAUR with DINO B/16 | 39.5 ± 0.6 | 29.1 ± 0.4 | 35.8 ± 0.3 | 29.4 ± 0.3 | 28.9 ± 0.4 | 26.3 ± 0.2 |
| VideoSAUR with DINOv2 B/14 [58] | 39.7 | 35.6 | 38.7 | 34.5 | 31.2 | 29.7 |

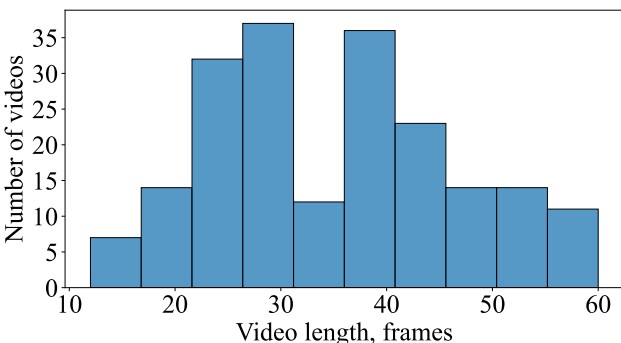

Figure B.1: Histogram of video lengths on the YT-VIS 2021 validation dataset.

## B.2 Optical Flow as Self-Supervised Target

The choice of the self-supervised target plays an important role in creating suitable inductive biases for object discovery and scene decomposition. As such, understanding the properties of the prediction target leading to an effective scene decomposition is crucial. The temporal feature similarity prediction proposed in this work combines two different inductive biases: high-level semantic information and motion information. To elucidate the significance of both types of bias for scene decomposition, we assess the performance of VideoSAUR with prediction targets that consist only of one of those biases.

In Table 2 and Table 3 in the main text, we compare predicting temporal similarities with predicting self-supervised features of the current frame. Such features only contain semantic information, but no information about motion. Depending on the dataset, including temporal information brings a small (YT-VIS) or large benefit (MOVi-C).

Subsequently, we study if motion cues alone (without the semantic information) are enough for successful scene decomposition. In particular, we compare self-supervised temporal similarity targets with (ground truth) optical flow targets (only motion information) on the MOVi-C and MOVi-E datasets. To this end, we train VideoSAUR by predicting optical flow targets, using a spatial broadcast decoder similar to SAVi [5] instead of the mixer decoder. All other components of VideoSAUR stay unchanged. The optical flow map is predicted at full image resolution ($128 \times 128$).

The results are presented in Table B.2. We train VideoSAUR with GT optical flow for the best potential performance from optical flow alone. Yet, even on the MOVi-C datasets favoring optical flow (no camera motion, no static objects), VideoSAUR with temporal feature similarities significantly outperforme optical flow (+10 FG-ARI). This disparity is even greater on the MOVi-E dataset, highlighting VideoSAUR's resilience to camera movements and static objects. Together, these results demonstrate that our temporal feature similarity targets, despite not requiring signals such as optical flow (which would need estimation in real-world scenarios like the YouTube-VIS dataset), excel over mere optical flow targets. We attribute this to the enriched semantic bias inherent to the self-supervised feature similarities.

Table B.2: Comparing VideoSAUR predicting temporal similarities to predicting ground truth optical flow on the MOVi-C and MOVi-E datasets. We report Video FG-ARI of a version of VideoSAUR with optical flow (both backward and forward) as well as the original VideoSAUR with temporal features similarity.

| VideoSAUR | MOVi-C | MOVi-E |
|---|---|---|
| w/ GT Optical Flow (backward) | 48.1 | 28.9 |
| w/ GT Optical Flow (forward) | 48.9 | 30.1 |
| w/ Temporal Similarities | 60.7 | 73.9 |

Table B.3: Extended ablation of VideoSAUR components on MOVi-E. We compare VideoSAUR model with different choices of the decoder (Mixer vs MLP used by DINOSAUR) and loss types (temporal similarity loss vs feature reconstruction).

| Decoder | Loss Type | FG-ARI | mBO |
|---|---|---|---|
| MLP | Feature Reconstruction | 68.6 | 27.6 |
| MLP | Temp. Feat. Sim. Prediction | 74.5 | 28.8 |
| Mixer | Feature Reconstruction | 62.3 | 20.6 |
| Mixer | Temp. Feat. Sim. Prediction | 74.1 | 34.1 |

## B.3 Comprehensive Ablation Study on the MOVi-E Dataset

In this section, we present the results of a comprehensive ablation study conducted on the MOVi-E dataset. The purpose of this study is to investigate the impact of two key factors: decoder choice (MLP vs. Mixer) and loss function selection. Our goal is to gain a deeper understanding of how these choices affect the performance of our method. The results are summarized in Table B.3.

The similarity loss is beneficial for both decoders, pushing the FG-ARI to approximately 74, as compared to 69 when using the feature reconstruction loss. Notably, the Mixer decoder significantly enhances the clarity of the object mask, with an improvement of $+5$ mBO. When combined, the similarity loss and Mixer consistently outshine the MLP decoder equipped with the feature reconstruction loss. These insights provide valuable auxiliary information to our main paper ablations (see Table 2 and Table 3), painting a more detailed picture of VideoSAUR's components and their respective performances.

## B.4 Stability of Mixer Decoder

As mentioned in Sec. 3.3 of the main text, we found that the mixer decoder sometimes exhibits training instabilities. For instance, Table B.4 shows that there is high variance over random seeds when training purely with feature reconstruction, i.e. some training runs fail to discover an object grouping. These instabilities manifest in slot masks that follow a Voronoi-like decomposition of the image. When adding the temporal similarity loss, the instabilities disappear.

We hypothesize this is because the mixer decoder has increased flexibility in how to model the image with slots (compared to the conventional mixture-based decoder), and thus more failure modes

Table B.4: Mixer decoder with smaller DINO features (S/16) on YT-VIS 2021 (mean $\pm$ standard dev., 3 seeds).

| Loss type | | Metric | |
|---|---|---|---|
| Feat. Rec. | Temp. Sim. | FG-ARI | mBO |
| ✓ | | 14.9 ± 12.0 | 12.9 ± 5.9 |
| ✓ | ✓ | 37.0 ± 3.5 | 29.1 ± 0.6 |

Table B.5: Loss ablation study on COCO dataset. Metrics are image-based ARI and mBO (mean, 3 seeds).

| Loss type | | Metric | |
|---|---|---|---|
| Feat. Rec. | Self-Sim. | FG-ARI | mBO$^i$ |
| ✓ | | 34.8 | 23.9 |
| | ✓ | 28.5 | 25.6 |
| ✓ | ✓ | 38.0 | 25.9 |

(non-object groupings) the model can "fall into" during training. Increasing the difficulty of the task by adding the temporal similarity loss makes these failure modes less viable: by putting more pressure on the slot bottleneck to encode information, object-based slot groupings are more efficient representations than alternative groupings.

However, we found that the mixer decoder with feature reconstruction does not show instabilities in *all* settings. For example, training with DINO ViT Base/8 features on MOVi-C or DINO Base/16 features on YouTube-VIS 2021 (Fig. 2 and Fig. 3 in the main text) is relatively stable. We attribute this to the increased task difficulty when predicting ViT "Base" features instead of ViT "Small" features (as in Table B.4). Once more [5, 6, 9, 26, 29], these findings demonstrate the central lesson of unsupervised object discovery: to be successful, the model needs to have sufficient inductive biases, whether they stem from the dataset, the decoder, the grouping module, or the training task.

### B.5 Image-Based Feature Similarity on COCO

In this section, we show that our proposed similarity loss can also be useful for image-based datasets, and thus is not restricted to just the video setting. Note that by setting the time-shift $k$ to $0$, the temporal similarities turn into a *self-similarities*, that is, the target similarities are computed by comparing features from the same image. The resulting similarity maps highlight semantically and spatially related patches and thus could be useful targets to discover objects.

To test this, we train the DINOSAUR method with ViT S/16 [9] with the self-similarity loss on the real-world COCO dataset (see Table B.5). Similar to the results from the time-shift analysis (see $k = 0$ in Fig. 6b), we find that using the self-similarity loss alone does not seem to carry enough signal to train the model and leads to degraded performance. However, combining the self-similarity loss with feature reconstruction shows significant improvement over using only feature reconstruction. Even though the targets contain the same information overall, different transformations of the original targets (e.g. their relative similarity) create different biases — a combination of these *different views into the targets* appears to be beneficial for object discovery.

### B.6 Sensitivity to Number of Slots During Evaluation

One of the noteworthy properties of slot attention-based models with randomly sampled initialization[5] is that the number of slots can be adjusted during inference. As we demonstrate in Fig. 6b, this is helpful for successfully transferring to datasets that have a different average number of objects per video. For this purpose, it is important to examine how stable the model's performance is if it is used with a different number of slots than during training. Thus, we evaluate our model (trained on YT-VIS 2021 with $k = 7$ slots) using a varying number of slots (from 1 to 12) and present the results in Fig. B.2. We observe that while using fewer slots steadily deteriorates the performance, our method performs relatively well with a larger number of slots. This suggests that our method is relatively robust to the usage of a larger number of slots than needed. This property is useful for object discovery in images where the number of objects is unknown.

### B.7 Effect of ViT Architecture on Final Performance

In addition to studying how the choice of self-superivsed method and ViT outputs affect the performance of VideoSAUR (see Fig. 6c and Fig. 6d in the main paper), we also explore the effect of the *scale of ViT architecture* on the performance. To this end, we compare VideoSAUR with DINO features of 2 different ViT sizes (Small and Base) and two different patch sizes ($16 \times 16$ and $8 \times 8$ resolutions) on the MOVi-C dataset (see Table B.6). The results are presented in Fig. B.3. We find that smaller patch size is important for sharper masks (measured by the mBO metric), while both larger architecture and smaller patch size are important for a better split of the scene to object masks (measured by the FG-ARI metric).

## C Architectural Details and Hyperparameters

Here we describe details about our model, its training and baselines that we use for comparison. We release our code at `https://github.com/martius-lab/videosaur`.

---

[5]This is in contrast to the fixed learned initialization used in unconditioned SAVi [5] and SAVi++[6].

## C.1 SlotMixer Decoder

We describe the SlotMixer decoder, and how we adapted it for 2D decoding. We keep the original terminology from Sajjadi et al. [13] for consistency. SlotMixer performs three steps for decoding: the *allocation step* assigns slots to spatial positions, the *mixing step* creates a slot mix for each spatial position, and the *render step* decodes the slot mix to the final output. See Fig. C.1 for an overview and pseudocode implementing the decoder.

**Allocation Step** This step takes as input the slots $s_t \in \mathbb{R}^{K \times M}$ and a learned positional embedding $p \in \mathbb{R}^{L \times M}$ and outputs a feature vector $f \in \mathbb{R}^{L \times M}$ using a cross-attention Transformer. In particular, this Transformer iterates several cross-attention operations (and applies residual two-layer MLPs) using the position embeddings as queries to attend into the set of slots, where the position embeddings are residually updated using values from the slots. Importantly, the position embeddings are processed independently of each other. We utilize pre-normalization and also apply a layer norm to the slots before feeding them into the Transformer. In contrast to Sajjadi et al. [13], we use a learned positional embedding initialized from a normal distribution instead of 3D encodings for the query rays, and do not apply a MLP to the positional embedding.

**Mixing Step** The mixing step is similar to a single-head attention step using the features $f$ as queries and the slots $s_t$ as keys, where the slots are averaged as the values to form the slot mix $m \in \mathbb{R}^{L \times M}$ that is used for decoding to the final output:

$$
\begin{aligned}
\mathbf{q} &= \mathrm{norm}(f)\,U_q & U_q &\in \mathbb{R}^{M \times M}, \\
\mathbf{k} &= \mathrm{norm}(s)\,U_k & U_k &\in \mathbb{R}^{M \times M}, \\
A &= \mathrm{softmax}\left(\mathbf{q}\mathbf{k}^\top / \sqrt{M}\right) & A &\in \mathbb{R}^{L \times K}, \\
m &= sA & m &\in \mathbb{R}^{L \times M}.
\end{aligned}
$$

**Render Step** The render step takes the slot mix $m \in \mathbb{R}^{L \times M}$, adds the positional embedding $p \in \mathbb{R}^{L \times M}$ and applies a MLP with ReLU activation independently to each position:

$$
y = \mathrm{MLP}(m + p).
$$

Instead of *adding* the positional embedding, we also explored concatenating it to the slot mix; we did not find large differences from doing so.

## C.2 ViT Encoders as Dense Features Extractors

The Vision Transformer (ViT) [43] architecture takes an input frame, denoted here as $x \in \mathbb{R}^{244 \times 224 \times 3}$, which is divided into a grid of non-overlapping contiguous patches of resolution $N \times N$. Each of these patches is then passed through a linear transformation to generate a set of patch feature embeddings, $h^0 \in \mathbb{R}^{L \times D}$.

The set of patch tokens is subsequently provided as input to a standard Transformer network. This Transformer network is comprised of a sequence of self-attention and feed-forward layers, alongside

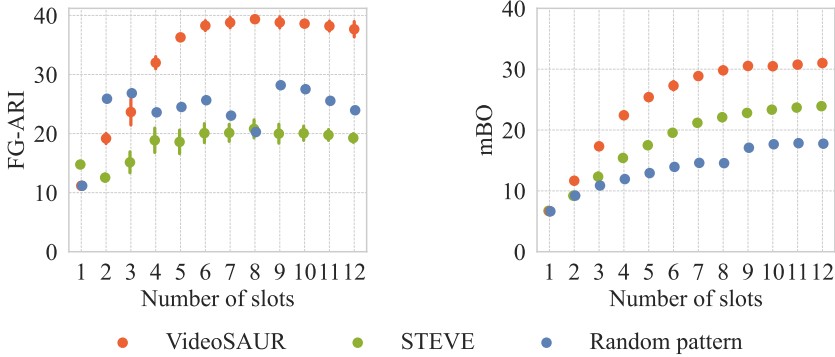

Figure B.2: Changing the number of slots during evaluation on the YT-VIS 2021 dataset (mean ± standard dev., 5 seeds).

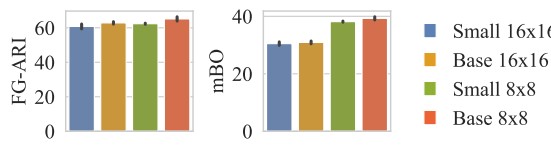

Figure B.3: Effect of ViT architecture choice and patch size on VideoSAUR training on the MOVi-C (mean ± standard dev., 3 seeds).

Table B.6: ViT networks configuration.

| Model | Patch Size | Dim | Heads | Tokens | Params |
|-------|-----------|-----|-------|--------|--------|
| Small | 16 | 384 | 6 | 196 | 21M |
| Small | 8 | 384 | 6 | 784 | 21M |
| Base | 16 | 768 | 12 | 196 | 85M |
| Base | 8 | 768 | 12 | 784 | 85M |

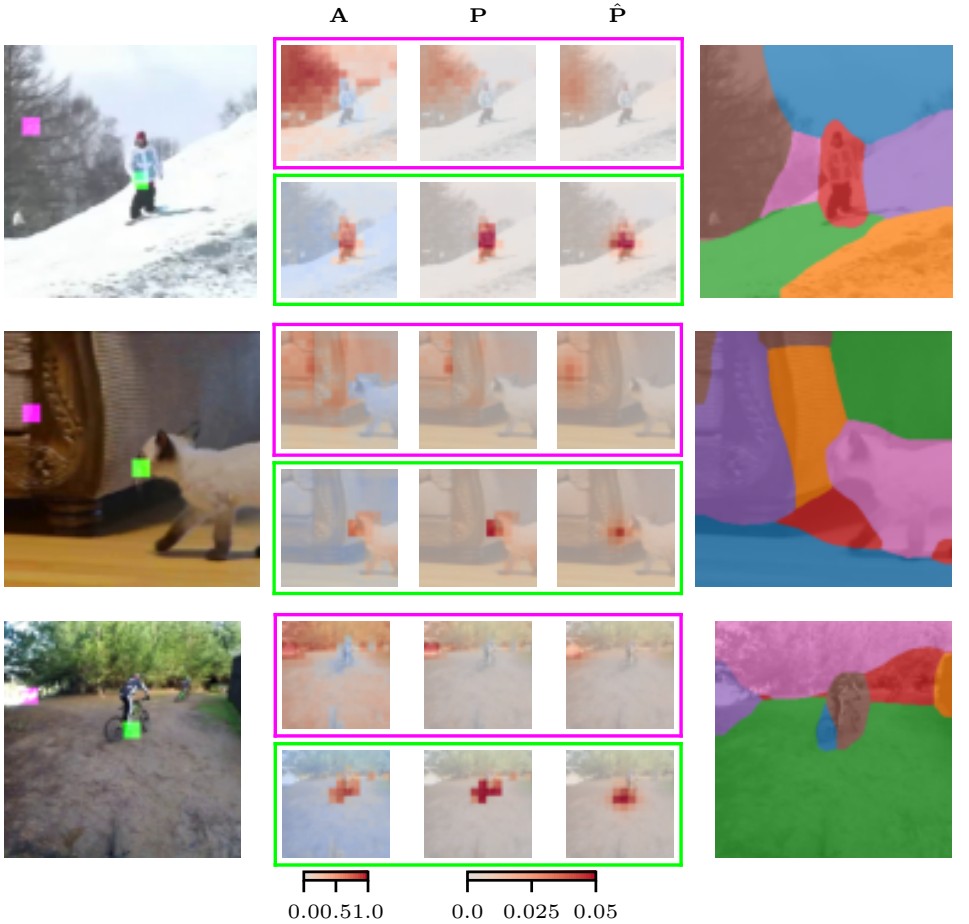

Figure B.4: Additional visualization of affinity matrix $\boldsymbol{A}$, transition probabilities $\boldsymbol{P}$ and decoder predictions of transition probabilities $\hat{\boldsymbol{P}}$ between patches (marked by purple and green) of the frame $\boldsymbol{x}_t$ and patches of the next frame $\boldsymbol{x}_{t+1}$ for YouTube-VIS 2021 validation videos. Red indicates maximum affinity/probability.

residual connections for each layer. These residual connections are important for letting each patch representation $\boldsymbol{h}^i$ keep a correspondence to the original image patch representation $\boldsymbol{h}^0$ and thus making the patch representation a dense (with the resolution $\sqrt{L} \times \sqrt{L}$) representation of the image. In the self-attention layers, the token representations are updated through an attention mechanism that takes into account the representations of all tokens:

$$
\begin{aligned}
[\mathbf{q}^i, \mathbf{k}^i, \mathbf{v}^i] &= \mathbf{h}^{i-1}\mathbf{U}_{qkv} & \mathbf{U}_{qkv} &\in \mathbb{R}^{D \times 3D} \\
\boldsymbol{A} &= \mathrm{softmax}\left(\mathbf{q}^i\mathbf{k}^{i\top}/\sqrt{D}\right) & \boldsymbol{A} &\in \mathbb{R}^{L \times L} \\
\mathbf{o}^i &= \boldsymbol{A}\mathbf{v}^i
\end{aligned}
$$

Here $\mathbf{q}^i, \mathbf{k}^i, \mathbf{v}^i, \mathbf{o}^i$ are queries, keys, values and outputs of the self-attention layer $i$. In contrast to DINOSAUR [9], which uses the outputs $\mathbf{o}^i$ as the target image representation, we use attention keys

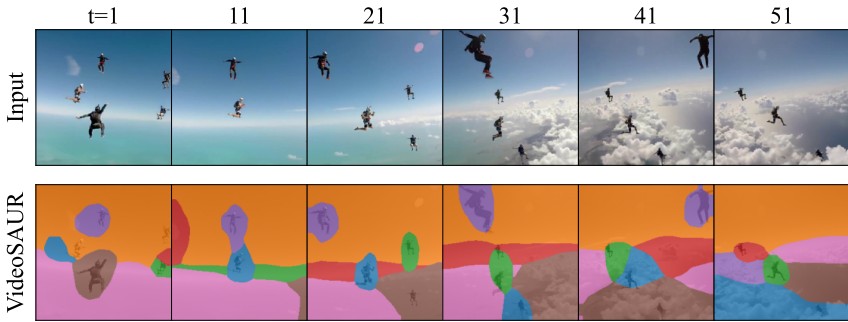

Figure B.5: Failure case for long video prediction. Note that the original videos in YouTube-VIS are resampled from the original 30 fps to 6 fps, thus the original length of the video is 250 frames. The slots are reassigned to the background, while small objects are not recognized.

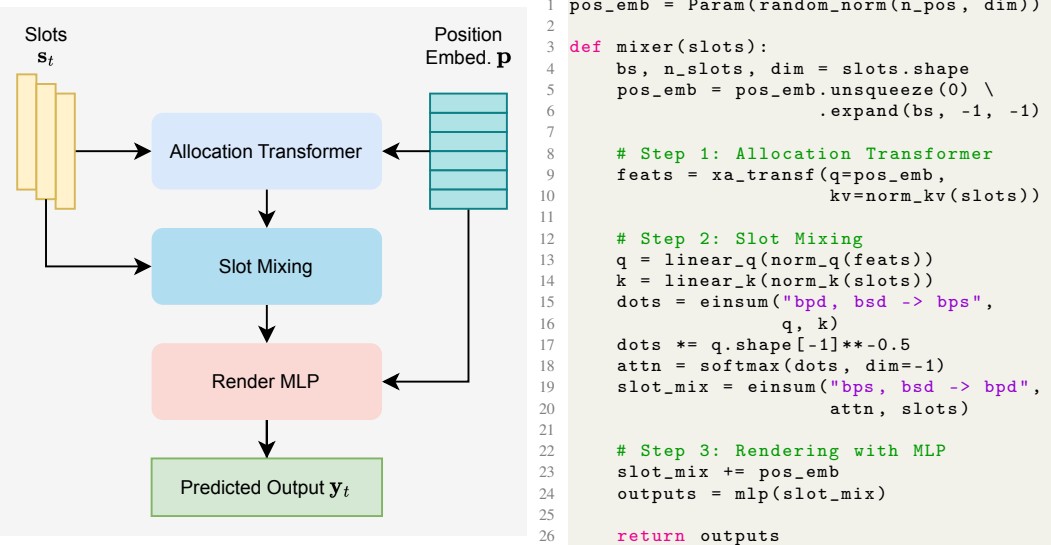

```
1  pos_emb = Param(random_norm(n_pos, dim))
2
3  def mixer(slots):
4      bs, n_slots, dim = slots.shape
5      pos_emb = pos_emb.unsqueeze(0) \
6                       .expand(bs, -1, -1)
7
8      # Step 1: Allocation Transformer
9      feats = xa_transf(q=pos_emb,
10                        kv=norm_kv(slots))
11
12     # Step 2: Slot Mixing
13     q = linear_q(norm_q(feats))
14     k = linear_k(norm_k(slots))
15     dots = einsum("bpd, bsd -> bps",
16                   q, k)
17     dots *= q.shape[-1]**-0.5
18     attn = softmax(dots, dim=-1)
19     slot_mix = einsum("bps, bsd -> bpd",
20                       attn, slots)
21
22     # Step 3: Rendering with MLP
23     slot_mix += pos_emb
24     outputs = mlp(slot_mix)
25
26     return outputs
```

Figure C.1: SlotMixer decoder. Left: SlotMixer performs three steps for decoding: the allocation transformer assigns slots $s_t$ to spatial positions $p$, the slot mixing step creates a slot mix for each spatial position, and the render MLP decodes the slot mix to the final output $y_t$. Right: PyTorch-like pseudocode for the SlotMixer decoder.

$k^i$ from the last self-attention layer of ViT as the dense image representation that is provided to the temporal similarity loss (see Fig. 6c for a detailed comparison of these representations). However, we still use the outputs $o^i$ as the input to the slot attention grouping module.

## C.3   Other Modules

We group the dense encoder features with a recurrent slot attention module similar to Singh et al. [7] and Kipf et al. [5]. First, we transform the original features with a two-layer MLP with an output dimension equal to the slot dimension. Second, we use a slot attention module initialized with randomly sampled slots to group the first frame features, while for subsequent frames, we initialize the slot attention module with the slots of the previous frame, additionally transformed with a predictor module. We use the GRU recurrent unit in the slot attention grouping, but not the residual MLP. Similar to SAVi [5] and STEVE [7], we use a one-layer transformer as the predictor module. In addition, we propose to decouple the number of Slot Attention iterations in the first frame and other frames of the video. This allows more iterations for the first frames (we use 3 iterations similar to image-based methods) and fewer iterations for the next frames where the initialization is much

Table C.1: Next-frame feature prediction with different decoders.

| | MOVi-C | | MOVi-E | | YT-VIS | |
|---|---|---|---|---|---|---|
| | FG-ARI | mBO | FG-ARI | mBO | FG-ARI | mBO |
| One decoder head | 44.6 | 23.5 | 61.3 | 22.1 | 33.4 | 24.6 |
| Two decoder heads | 47.2 | 24.7 | 62.9 | 24.0 | 37.9 | 27.3 |

better (we use 2 iterations). For computational reasons, we were training on relatively short 4-frame segments of original videos, i.e. $T = 4$.

## C.4 Next-Frame Feature Prediction Details

In this part, we cover implementation details for the next-frame feature prediction ablation presented in Table 2 and in Table 3. Reconstructing frame features from the current and next frame simultaneously with a single decoder is problematic because the decoder masks that are used for evaluation would be in reference to both the current and next frame. One way to overcome this problem is by using two decoder heads: $d_{current}$ for the current frame and $d_{next}$ for the next frame. Each head produces its own predictions and masks. In this case, masks from the $d_{current}$ head can be used for evaluation. While more powerful, this approach also requires more memory and is slower than standard setting with only one head. In our experiments, we confirm that the version with two different Mixer decoders performs better than simultaneous reconstruction with one decoder (see Table C.1). We use this better version for our comparisons even though it is heavier than our method which needs only one decoder.

## C.5 Baselines

**Block Pattern**    The block pattern baseline serves to show metrics for a trivial decomposition of the video into regular blocks. It is intended to show the difficulty of the dataset and how much object-centric methods improve upon such a trivial solution, as the metrics values could be difficult to interpret without further calibration. To this end, we are splitting the video to $k$ spatial blocks consistently for all frames of the video, similar to how Seitzer et al. [9] are splitting images into regular blocks.

**SAVi**    We reimplement SAVi [5] close to the official implementation[6]. In particular, we use the SAVi-L architecture for all experiments. This corresponds to the version using a ResNet-34 encoder as described by Kipf et al. [5], and a CNN broadcast mixture decoder with 4 layers. We apply the unconditional version of SAVi, using a fixed learned slot initialization instead. We train the model for 200 000 steps on all datasets, with a batch size of 64, using image reconstruction as the training signal. For training, we use videos with 4 frames, with a single slot attention step per frame.

**STEVE**    We reimplement STEVE [7] close to the official implementation[7]. We use the proposed configuration for the MOVi datasets and only change the number of slots to 11 for MOVi-C, 15 for MOVi-E and 7 for YT-VIS 2021. STEVE trains a dVAE [50] on the video frames to extract a discrete latent code that is used as the reconstruction target. STEVE uses a CNN encoder with 4 layers and a Transformer decoder with 8 layers. We train the model for 200 000 steps for MOVi-C and YT-VIS datasets and for 100 000 steps for MOVi-E datasets which resulted in optimal performance on this dataset, with a batch size of 24. Like Singh et al. [7], for training, we use videos with 3 frames, with two slot attention steps per frame.

## C.6 Hyperparameters

The full list of the hyperparameters used to train VideoSAUR on MOVi-C, -D, -E, and YouTube-VIS 2021 datasets are presented in Table C.2. Most of the hyperparameters are similar to previous methods, while new ones (such as softmax temperature $\alpha$ and time shift $k$) are optimized using a grid search.

---

[6] https://github.com/google-research/slot-attention-video/
[7] https://github.com/singhgautam/steve

Table C.2: Hyperparameters of VideoSAUR for the main results on MOVi-C, MOVi-E, and YouTube-VIS 2021 datasets.

| Dataset | | **MOVi-C** | **MOVi-E** | **YouTube-VIS** |
|---|---|---|---|---|
| Training Steps | | 100k | 100k | 100k |
| Batch Size | | 128 | 128 | 128 |
| Training Segment Length | | 4 | 4 | 4 |
| LR Warmup Steps | | 2500 | 2500 | 2500 |
| Optimizer | | Adam | Adam | Adam |
| Peak LR | | 0.0004 | 0.0004 | 0.0004 |
| Exp. Decay | | 100k | 100k | 100k |
| ViT Architecture | | ViT Base | ViT Base | ViT Base |
| Patch Size | | 8 | 8 | 16 |
| Feature Dim. $D_{\text{feat}}$ | | 768 | 768 | 768 |
| Gradient Norm Clipping | | 0.05 | 0.05 | 0.05 |
| Image/Crop Size | | 224 | 224 | 224 |
| Cropping Strategy | | Full | Full | Rand. Cent. Crop |
| Augmentations | | – | – | Rand. Hori. Flip |
| Image Tokens | | 784 | 784 | 196 |
| Slot Attention | Slots | 11 | 15 | 7 |
| | Iterations (first / other frames) | 3/2 | 3/2 | 3/2 |
| | Slot Dim. $D_{\text{slots}}$ | 128 | 128 | 64 |
| Predictor | Type | Transformer | Transformer | Transformer |
| | Layers | 1 | 1 | 1 |
| | Heads | 4 | 4 | 4 |
| Decoder | Type | Mixer | Mixer | Mixer |
| | Allocator Transformer Layers | 2 | 2 | 3 |
| | Allocator Transformer Heads | 4 | 4 | 4 |
| | Renderer MLP Layers | 4 | 4 | 3 |
| | Renderer MLP Hidden Dim. | 1024 | 1024 | 1024 |
| Loss | Softmax Temperature $\tau$ | 0.075 | 0.075 | 0.25 |
| | Time-shift $k$ | 1 | 1 | 1 |
| | Feature reconstruction weight $\alpha$ | – | – | 0.1 |

## C.7 Compute Requirements

We used a cluster of A100 GPUs (with 40 and 80 Gb memory) for running the experiments. A single training run (100k steps) of VideoSAUR equipped with DINO B/16 with a batch size of 128 takes roughly 18 hours on one A100 GPU with 40 GB memory. We use 5 seeds for the final results and 3 seeds for other experiments. Overall, we estimate the total compute spend on the whole project including training of all baselines and the method development (including dead ends) to be around 800-1000 GPU days (this is a rough estimate obtained from our cluster usage).

## D  Dataset Details

In this section, we provide details about the datasets used in this work. See Table D.1 for an overview.

**MOVi datasets**    For MOVi-C, MOVi-D and MOVi-E, we use the standard training and test splits provided in the respective releases of MOVi datasets: 9750 training videos and 250 validation videos. For a fair comparison, all the methods are trained on images with the same initial resolution $128 \times 128$. Consistent with previous work [5–7, 9], we utilize the validation split of the MOVi dataset for our evaluations.

**YouTube-VIS datasets**    The YouTube-VIS datasets [14, 46] are benchmarks originally used for *supervised* video instance segmentation. The video instance segmentation task involves segmenting, tracking and classifying instances throughout a video. It is a challenging dataset because it contains

Table D.1: Overview of datasets used in this work. For the training process, we solely utilize images or videos derived from the relevant datasets, with no reliance on labels. To generate central crops, we initially resize the mask so that its shorter dimension is 224 pixels. Subsequently, we extract the most centrally located crop, maintaining a size of 224 by 224 pixels.

| Dataset | Videos | Images | Description | Citation |
|---|---|---|---|---|
| MOVi-C, -D, -E | 9 750 | – | Train split videos | Greff et al. [12] |
| MOVi-C, -D, -E validation | 250 | – | Val. split w. instance segm. labels | Greff et al. [12] |
| COCO 2017 | – | 118 287 | Train split | Lin et al. [48] |
| COCO 2017 validation | – | 5 000 | Val split w. instance segm. labels | Lin et al. [48] |
| YouTube-VIS 2021 | 2785 | – | Part of train split videos | Yang et al. [14] |
| YouTube-VIS 2021 | 210 | – | Part of train split w. instance segm. labels | Yang et al. [14] |
| YouTube-VIS 2019 | 200 | – | Part of train split w. instance segm. labels | Yang et al. [46] |
| DAVIS-2017-Unsupervised | 30 | – | Val. split w. instance segm. labels | Pont-Tuset et al. [47] |

various different classes of objects and the complexities of real-world video dynamics. To the best of our knowledge, this is the first work attempting *unsupervised* instance segmentation on YouTube-VIS.

There are two different versions of this dataset: YouTube-VIS 2019 and YouTube-VIS 2021. The YouTube-VIS 2019 dataset is mainly derived from the Video Object Segmentation (VOS) dataset. It has a limited number of individual instances (an average of 1.7 per video for the training set), and the categories of instances in the same video are usually different. In contrast, the 2021 edition of YouTube-VIS incorporates a higher quantity of objects with more difficult trajectories (average 3.4 per video for the additional videos in the train set) and thus are more interesting for object-centric learning. We sample both training and validation frames with a rate of 6 frames per second (each 5th frame of the original videos with 30 *fps*).

As the original validation dataset is not publicly available, and the evaluation server does not compute the object-centric metrics we need, we split the original training set into two parts (210 videos for validation and the other videos for training). Overall, we use 2775 videos for training, and 210 videos additionally added in YT-VIS 2021 train for validation.

**DAVIS dataset** The DAVIS-2017-Unsupervised dataset [47] is used for video object segmentation and also contains videos with multiple objects per video (average is 1.97 for 30 validation videos). As those videos are not related to YouTube-VIS videos, this dataset is useful for the evaluation of transfer abilities for real-world object-centric learning algorithms.

**COCO dataset** To test the properties of our proposed similarity loss for image-based object-centric learning we use the COCO dataset [48]. Similar to Seitzer et al. [9], we use *118287* training images (without labels) to train and 5 000 validation images (with labels) for performance evaluation.

# E Additional Examples

We include additional example predictions of our model:

- Figure E.1: comparing VideoSAUR to STEVE on Youtube-VIS 2021.
- Figure E.2: comparing VideoSAUR to STEVE on MOVi-C.
- Figure E.3: comparing VideoSAUR to STEVE on MOVi-E.
- Figure E.4: comparing feature reconstruction and temporal similarity loss on MOVi-C.
- Figure E.5: long-term video predictions on Youtube-VIS 2021 (VideoSAUR with DINO B/16 features).
- Figure E.6: long-term video predictions on Youtube-VIS 2021 (VideoSAUR with DINOv2 B/14 features).

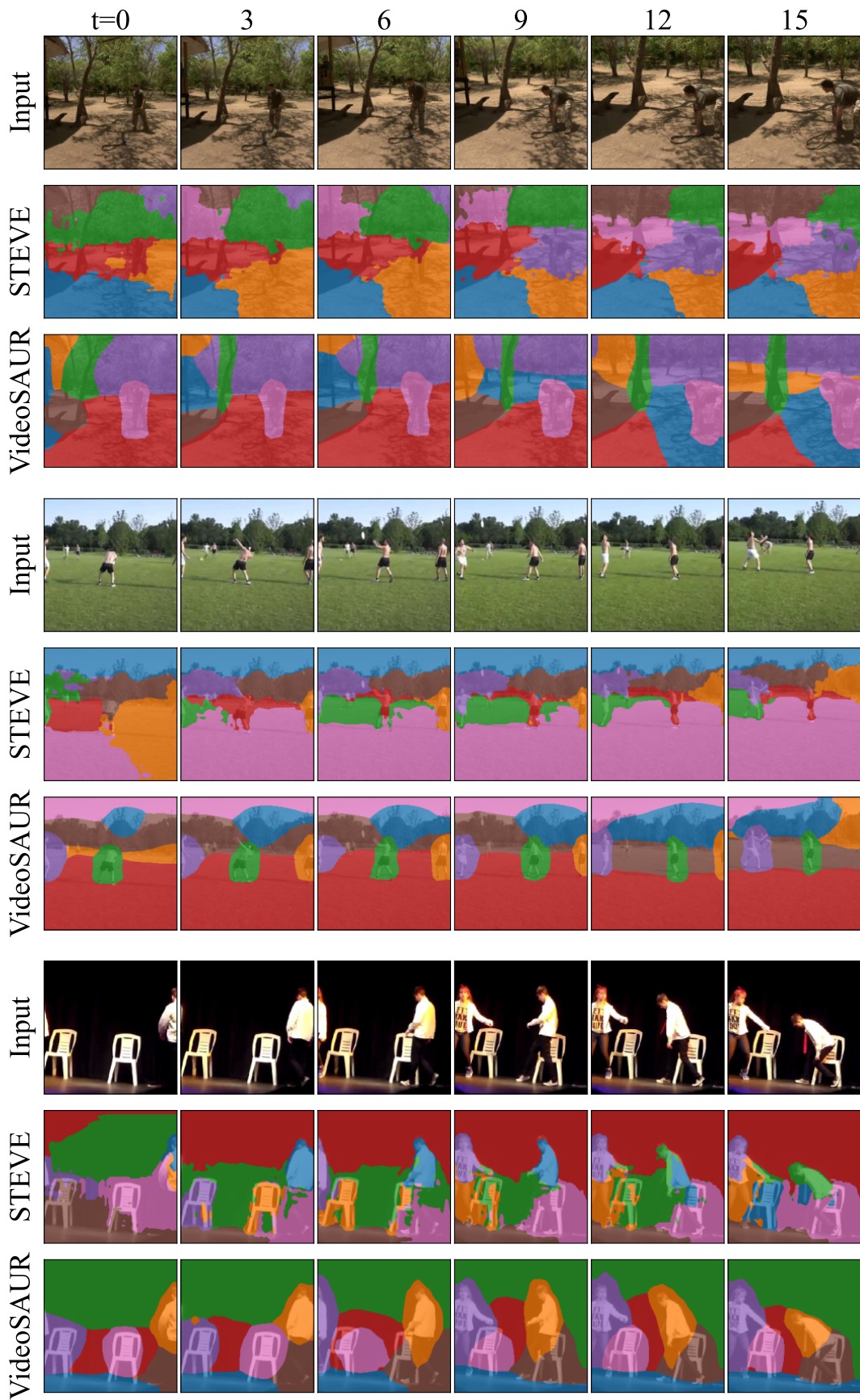

Figure E.1: Additional examples on YouTube-VIS 2021.

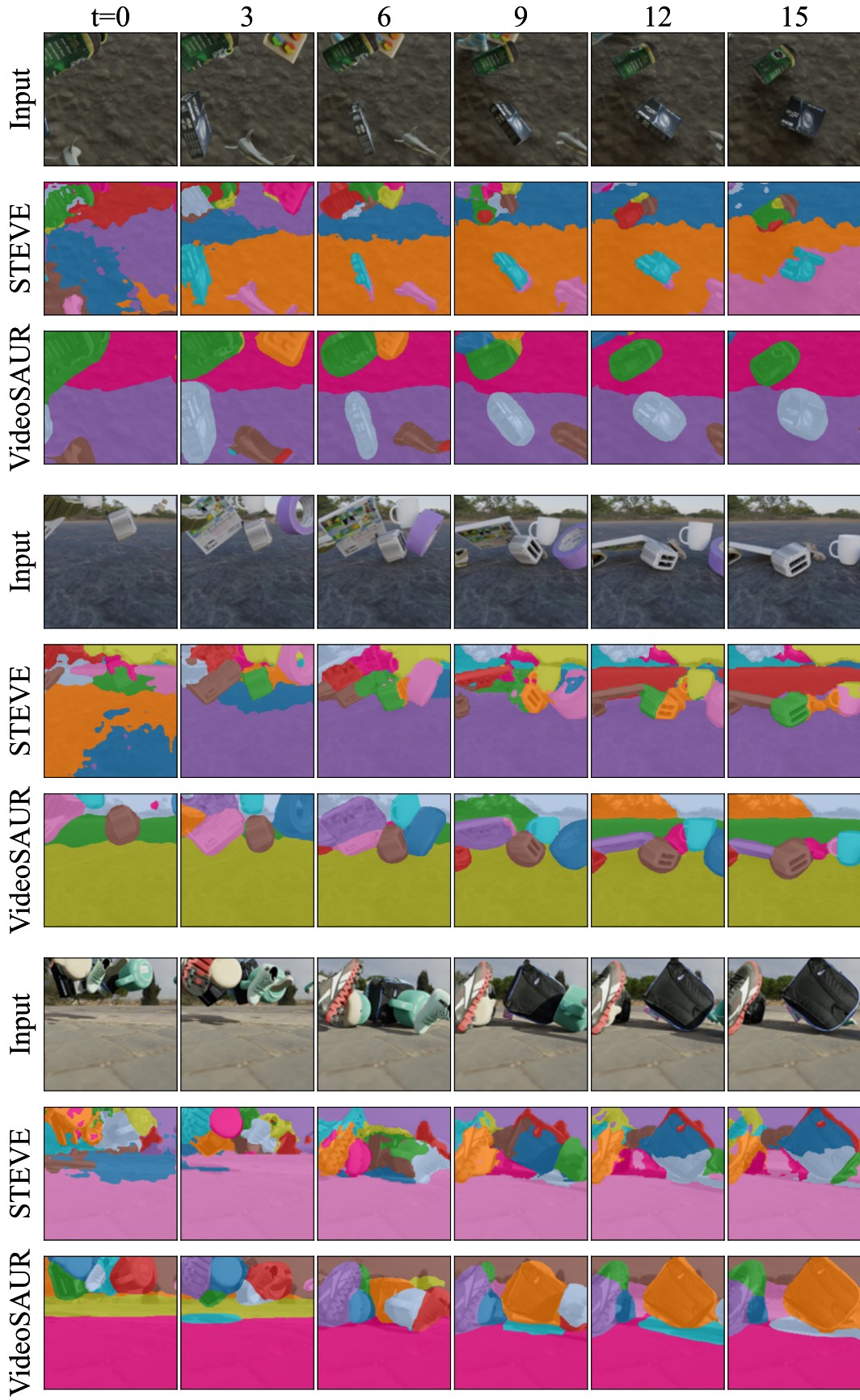

Figure E.2: Additional examples on MOVi-C.

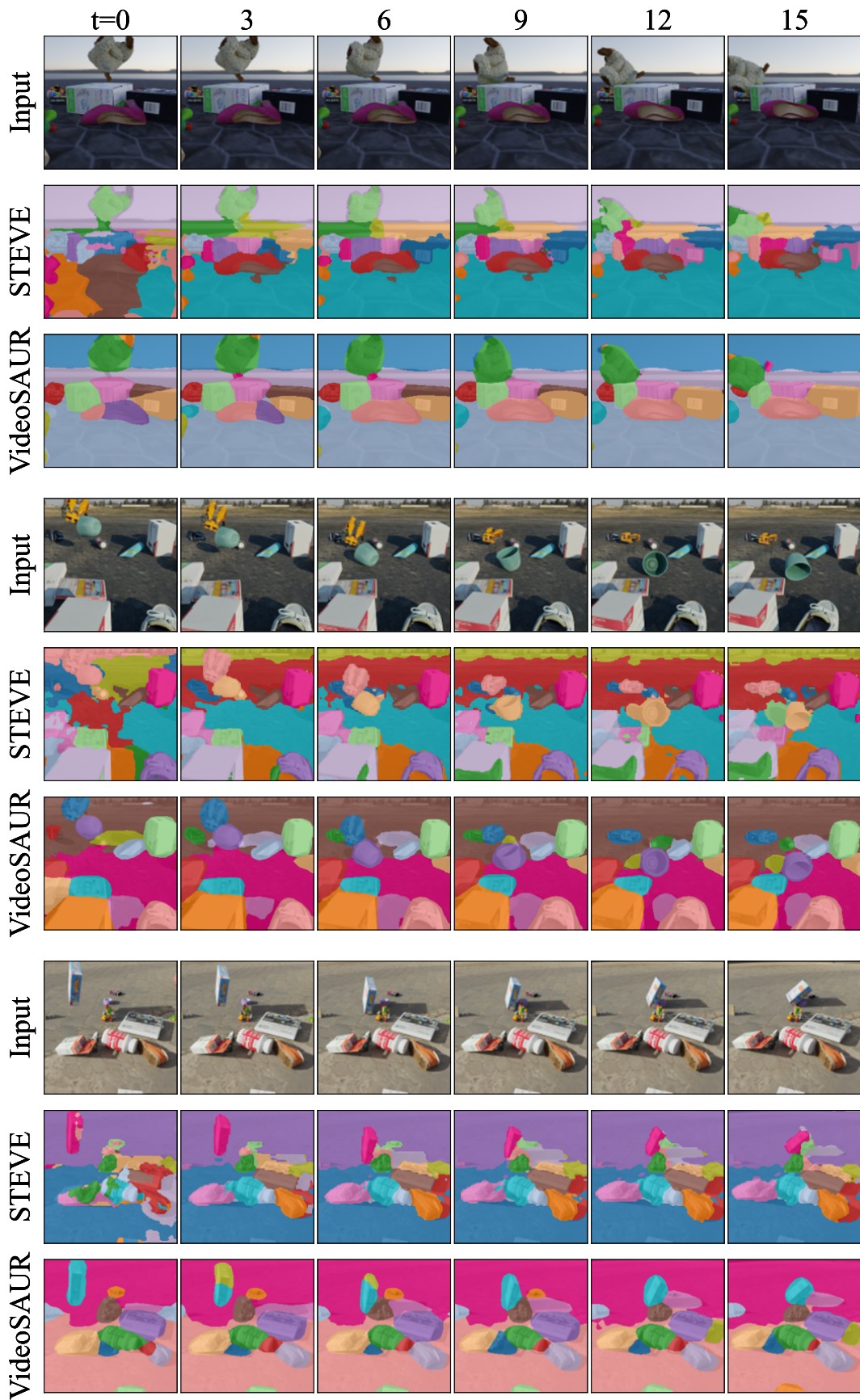

Figure E.3: Additional examples on MOVi-E.

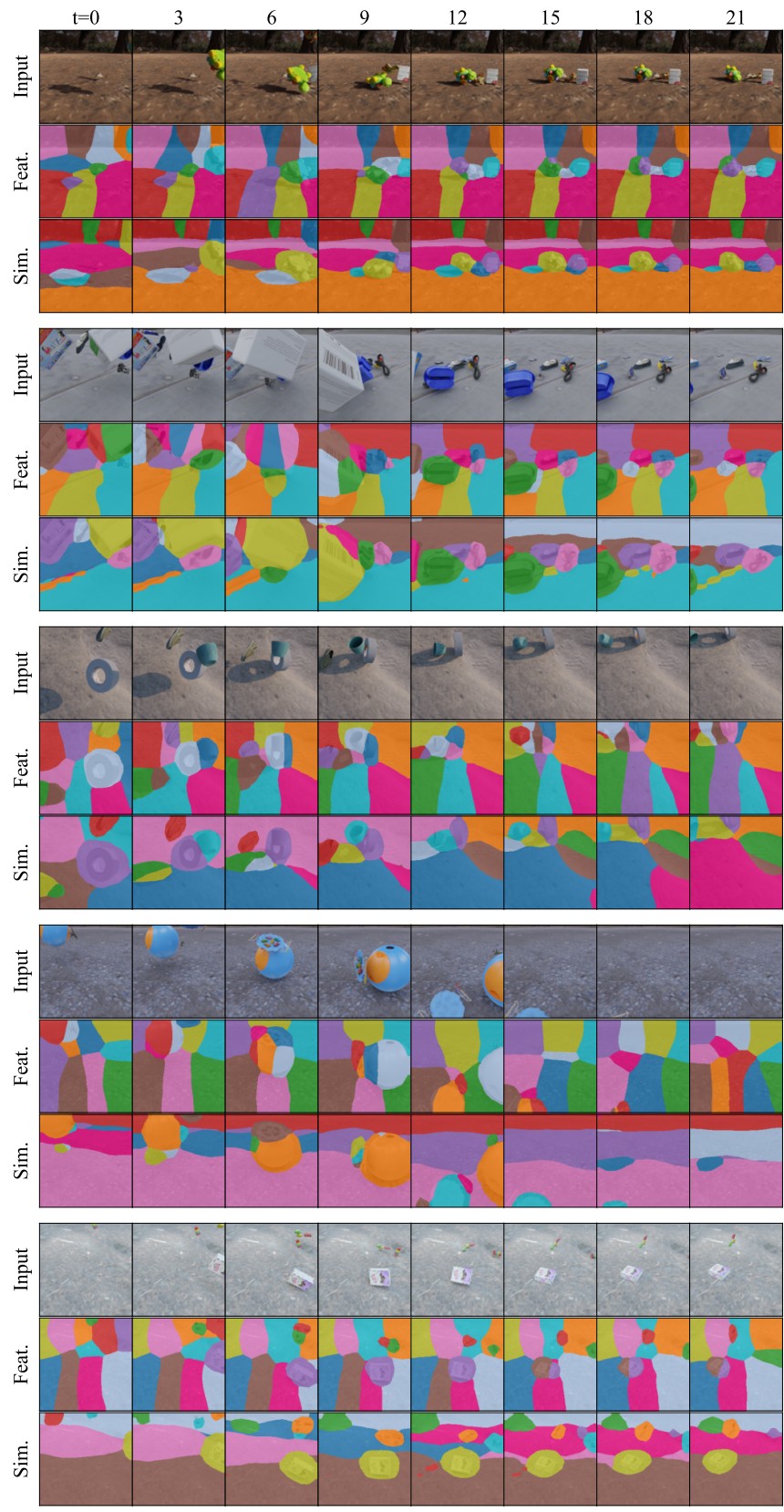

Figure E.4: Difference between VideoSAUR train with feature reconstruction and temporal feature similarity losses on MOVi-C. We show videos with larger differences in performance between the two methods.

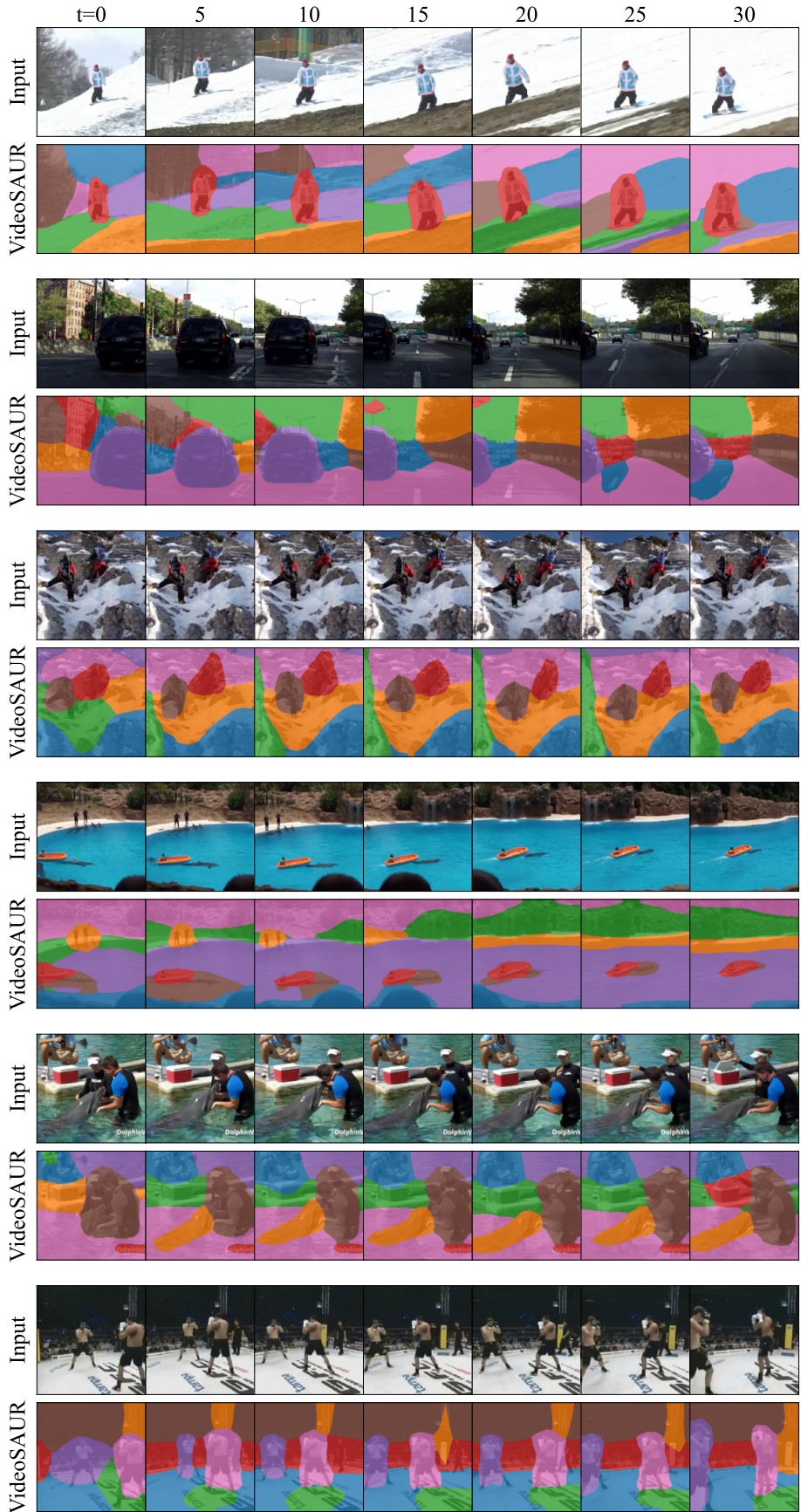

Figure E.5: Prediction of VideoSAUR with DINO B/16 features on longer videos from YouTube-VIS 2021.

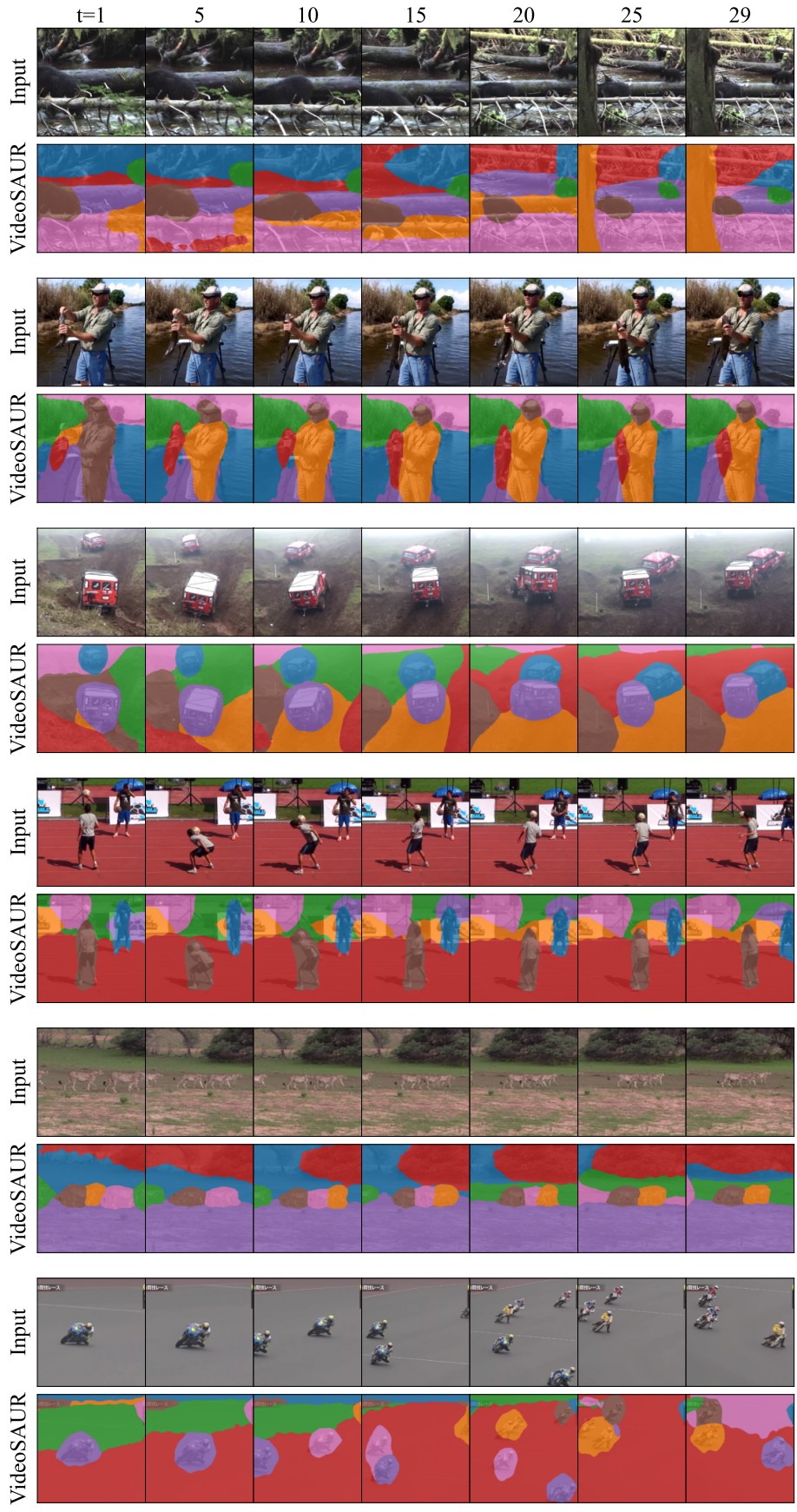

Figure E.6: Prediction of VideoSAUR with DINOv2 B/14 features on longer videos from YouTube-VIS 2021.

