# OpenReview forum: "Object-Centric Learning for Real-World Videos by Predicting Temporal Feature Similarities"
_NeurIPS.cc/2023/Conference — NeurIPS 2023 poster_

### Official Review · Reviewer_XQ9f · 2023-06-30

**Soundness:** 3 good
**Presentation:** 2 fair
**Contribution:** 2 fair
**Rating:** 5
**Confidence:** 5

**Summary:**

The authors combine several recent approaches for object-centric learning. In particular, they use the overall framework which is a combination of SAVi with a slot mixer decoder from [13]. The main objective is optical flow prediction, like in SAVi, but they combine it with DINO feature reconstruction from Dinosaur [9]. They also adopt a pre-trained and frozen DINO backbone from Dinosaur. The only novel component in this framework is that they estimate the ground truth optical flow using the model itself by computing similarity between encoder features in consecutive frames. The way this is done is also adopted from prior work (CRW [41]). Another delta with respect to SAVi is that they predict the future optical flow t -> t + 1 at time time t, in contrast to the standard flow estimation objective (t -1 -> t), but the ablations don't show whether this makes a difference.

In the experimental evaluation on MOVi-C and MOVi-E they outperform some baselines but do not compare to the sota methods [9, 36]. They additionally report results on the real world YouTube-VIS benchmark where they outperform the same baselines, but the results are low for all the methods. Ablation demonstrates that on the real videos most of the performance is achieved by the Dino feature reconstruction objective from [9] and the flow prediction objective is fairly ineffective in the real world.

**Strengths:**

The proposed approach seems sound.

A relatively thorough ablation study is provided.

The paper is relatively well written in terms of the language (but not the notation or organization).

State-of-the-art results are reported on the MOVi-E benchmark.

**Weaknesses:**

The proposed approach is merely a combination of several recent techniques for object centric learning. This wouldn't necessarily be a problem in itself, but the authors are not being fully upfront about it. In particularly, they do not explain that their 'feature similarity loss' is equivalent to optical flow prediction from SAVi (though they do show in Figure 3 that the learned feature similarities are effectively optical flow). In any case, the novelty of the proposed framework is minimal.

The authors do not compare to the state-of-the-art video-based and image-based approaches ([36] and [9]) in the main paper. They do compare to [9] on MOVi in the supplementary, but not on YouTube-VIS where that method is expected to be most effective (see ablation in Table 3). According to results on MOVi-C from the supplementary of [36] it outperforms the proposed approach. Thus it only seems to show  top performance on MOVi-E.

The results on YouTube-VIS are low for all the reported approaches. The numbers seem to indicate that the proposed methods simply fails less catastrophically than the baselines in the real world.

Although a fairly thorough ablation study is provided it is not consistent enough to understand the main reason why this method outperforms prior work on MOVi-E. For example, the effect of replacing the decoder in [9] is not evaluated. At least the full version of the proposed approach with the decoder from [9] should be reported. Another important baseline is using actual optical flow (ground truth or estimated with a pre-trained model) in place of P_{t, t+1}.

The authors claim to show generalization of the proposed approach by evaluating a model trained on YouTube-VIS on DAVIS, but these experiments are not indicative of anything. It is well known in the video segmentation community that the data distributions of these two datasets are very close and models trained on YouTube-VOS/VIS perform very well on DAVIS without fine-tuning (fine-tuning brings only marginal improvements). It would be better to replace these results with a proper comparison to the state-of-the-art and a more thorough ablation analysis.

There are still quite a few issues with the presentation. In particular, the notation is not consistent: for example y_t denotes the similarity matrix in Eq. 5 but previously it is stated that y_t is the output of the model which is used both for reconstruction and for flow prediction. Later a distinction is made between y_sim and y_rec but until that point it's hard to understand what the authors are talking about. In addition, Figure 2 is never mentioned in the text and Figure 1 is only mentioned on page 5. Captions are also not self-sufficient to parse these figures.

In addition, some of the claims made in the paper are inaccurate. In particular, the authors claim that 'the approaches that are based on optical-flow/motion masks are not self-supervised because they may be hampered by the availability of labels'. This is not true, because self-supervised flow/motion segmentation estimation methods exist.

**Questions:**

Report results of [9] on YouTube-VIS.

Report a full ablation on MOVi-E starting from Dinosaur and arriving to the proposed approach (ablate both the decoder, the new objective and all the changed hyper-parameters).

Report results for the variant of the mode that uses gt/estimated optical flow directly in the objective.

Ablate whether predicting flow into the future makes a difference compared to the variant of the objective used in SAVi.

**Limitations:**

Limitations are addressed in the manuscript.

---

> ### Author Rebuttal · Authors · 2023-08-09
>
> Thank you for the constructive suggestions to improve our paper.
> > The proposed approach is merely a combination of several recent techniques for object centric learning. […] The novelty of the proposed framework is minimal
>
> All in all, we find this statement not a fair characterization of our contributions. We summarize them in the general response as supported by the other reviewers. In the following discussion the points will be made more specific.
> > 'Feature similarity loss' is equivalent to optical flow prediction from SAVi.
>
> Our feature similarity loss is *different from SAVi’s optical flow prediction conceptually, mathematically, and in terms of the resulting model performance.*
> - Conceptually: for the feature similarity loss, a successful prediction requires to predict where *all semantically similar patches* are located in the next frame (see e.g. Fig. 3). In contrast, the optical flow target only contains information on how much each pixel moves independently from each other. Thus, the temporal feature similarity target is still meaningful for non-moving objects, whereas the optical flow target is trivial.
> - Mathematically: for optical flow prediction, SAVi uses MSE against the RGB image, leading to a point prediction. For feature similarity, we use the cross entropy loss against the patch similarity distribution, predicting a categorical distribution. This allows modeling uncertainty especially when similar patches are prevalent or when objects enter or leave the scene.
> - Performance: following the reviewer's feedback, we compared optical flow targets to VideoSAUR's feature similarity targets (**Table R3**). We trained VideoSAUR with GT optical flow for the best potential performance from optical flow alone. Yet, even on the MOVi-C datasets favoring optical flow, VideoSAUR with feature similarity significantly outperformed optical flow (+10 FG-ARI). This disparity is greater in the MOVi-E dataset, highlighting VideoSAUR's resilience to static objects and camera movement. We show that our temporal feature similarity approach that doesn't need signals like GT optical flow. Additionally, we also don’t need extra datasets (**Table R4**) to perform well.
>
> These new results now lead to a clearer picture of our contribution. We will adapt the text accordingly and rephrase the relation and comparison to optical flow in the paper.
> > … similarity between encoder features in consecutive frames is also adopted from prior work (CRW [41]).
>
> We clarify that both the loss function and the overall learning setup in CRW are different from our work in our response to reviewer L1X2.
> > Report results of [9] on YouTube-VIS.
>
> Thank you for the suggestion. To compare our method with DINOSAUR on the YouTube-VIS dataset, we train DINOSAUR and assess both approaches, along with STEVE and SAVi baselines, using image-based FG-ARI. **Table R1** displays the results. VideoSAUR outperforms DINOSAUR (+4 FG-ARI) and also surpasses video-based STEVE and SAVi methods, as shown in Table 1 of the main paper. This underscores the benefit of our temporal similarity loss over mere feature reconstruction.
> > […] why this method outperforms prior work on MOVi-E.
>
> We conducted an in-depth ablation study on the MOVi-E dataset, assessing the impact of the decoder (DINOSAUR MLP vs Mixer) and loss function choices. As shown in **Table R2**, using similarity loss improves FG-ARI for both decoders and reaches around 74 FG-ARI (vs 69 for feature reconstruction loss). The Mixer decoder enhances object mask sharpness by +5 mBO. Consequently, using similarity loss and Mixer jointly outperforms the MLP decoder with feature reconstruction loss. These findings supplement our main paper, offering a clearer insight into VideoSAUR's components and performance.
> > The results on YouTube-VIS are low for all the reported approaches. […] simply fails less catastrophically than the baselines in the real world.
>
> We respectfully disagree:
> - Qualitatively: we provide extensive visualizations (Figure 4, Figure E.1, Figure E.5) and on our website, demonstrating the successful operation on many different YT-VIS examples, which waves “simply fails […] catastrophically”.
> - Quantitatively: the regular pattern baseline is a good estimate of catastrophic failure. While both SAVi and STEVE perform worse or similar to that, VideoSAUR performs *twice* better, showing a clear and significant signal of “working”.
>
> Being the first to attempt the task of fully unsupervised video-object discovery on YouTube-VIS, we present a meaningful step forward and a large improvement over all compared baselines.
> > […] evaluating a model trained on YouTube-VIS on DAVIS […] are not indicative of anything.
>
> The distribution of objects is different in DAVIS and YouTube-VIS (e.g. number of objects). The transfer shows the flexibility of the object-centric model during inference. We note that generalizing results from supervised methods might be highly misleading. Nevertheless, we would consider moving this part to the supplementary.
> > Inaccurate claims
>
> The quoted passage by the reviewer differs from our actual statement in L150-L152: “Another advantage of our loss is that it is fully self-supervised, whereas approaches based on optical flow or motion masks may be hampered by the availability of labels. This is of particular importance for in-the-wild video, where motion estimation is challenging.” It highlights our loss advantage, especially when optical flow/motion estimation is tough in in-the-wild videos.
> We acknowledge the phrasing could be clearer and suggest: “Methods based on optical flow or motion masks can struggle with the need for accurate flow/motion mask labels (GT or estimated one), unlike our loss which doesn't require them.”
> > Notation and captions
>
> Thank you. We will clarify the notations for the decoder and improve the descriptions of the figures. We also note that the figures were referenced earlier in the Introduction and Section 3 using “Fig.”

---

> > ### Comment · Reviewer_XQ9f · 2023-08-13
> > **Re:re**
> >
> > I thank the authors for their detailed response. It did address some of my concerns. In particular, the new results demonstrate that the proposed objective is indeed markedly different from optical flow prediction. However, the reasons behind its effectiveness are still not entirely clear and require further analysis. For examples, it would be informative if the authors could provide additional results requested by reviewer L1X2.

---

> > > ### Author Response · Authors · 2023-08-16
> > > **Additional experimenal results requested by reviewer L1X2**
> > >
> > > We are happy that we could address your concerns. Regarding the effectiveness of our temporal similarity loss, we now ran the experiment suggested by reviewer L1X2 (described in details in a comment to this reviewer). To summarize, we find that optimizing next and current frame feature reconstruction improves performance over just optimizing current frame feature reconstruction, but that optimizing our proposed objective **brings clear additional improvements** on all datasets.
> > >
> > > This demonstrates that the effectiveness of our approach does not just stem from predicting the future, but also from the specific form of our similarity objective. We hope that we could address your final concerns with this experiment. If so, we would politely ask to adjust the score accordingly.

---

> > > > ### Comment · Reviewer_XQ9f · 2023-08-20
> > > > **Re:re**
> > > >
> > > > I thank the authors for providing the requested results. Overall, I find the response satisfactory. Fixing all the issues identified in the review process will require a major revision of the paper, but I trust that the authors will be able to do it without a resubmission. The most important changes include reporting the actual state-of-the-art methods in video and image settings in the main paper, clearly establishing a relationship between the proposed approach and prior work which is based on optical flow prediction and providing a detailed ablation that demonstrates the reasons behind the success of the proposed objective. I will update the score accordingly.

---

> ### Comment · Area_Chair_NSKm · 2023-08-20
> **Reminder from AC**
>
> Dear Reviewer
>
> Could you please check the rebuttal, if you have further concerns ?
>
> Best,
> AC

---

### Official Review · Reviewer_L1X2 · 2023-07-01

**Soundness:** 3 good
**Presentation:** 4 excellent
**Contribution:** 3 good
**Rating:** 6
**Confidence:** 5

**Summary:**

The paper aims to learn an effective object-centric feature representation for video segmentation. They build their method upon the slot attention-based framework and self-supervised ViT encoders (DINO), and propose a temporal feature similarity loss for object-centric learning. Their proposed method yields the best performance on several synthetic and real-world datasets, for both unsupervised object discovery and downstream video object segmentation tasks.

**Strengths:**

+ The paper is well-organized and easy to follow
+ The proposed temporal feature matching loss is sound and effective
+ The experimental design is comprehensive and promising
+ The proposed VideoSAUR model yields state-of-the-art performance on several datasets

**Weaknesses:**

I have two minor concerns regarding the method and experimental design:
- One of my concerns is the use of the pre-trained ViT encoder. The majority of the experiments are evaluated with a DINO encoder, however, DINO is pre-trained on ImageNet and DINO itself is "object-centric" (see https://github.com/facebookresearch/dino/). Though the authors have already clarified that they aim to bridge the gap between the pre-trained feature encoder and real-world video object discovery, it is still interesting to see if the proposed loss is useful *without* prior knowledge. It would be useful to add an experiment that uses a self-supervised ViT ***which is only pre-trained on the target dataset, e.g. MOVi-E***. Any self-supervised pre-training (DINO, MAE, MSN or MOCO) should be fine. This additional experiment could also give a fair comparison to other models as they, at least STEVE and SAVI, did not get access to extra data for pre-training.
- Novelty is the other concern. Though the authors claim the temporal feature matching loss is novel, the overall idea is adapted from contrastive random walk [41]. The proposed normalization function (Eq. 4) is different from the original one, but ignoring the negative-scored patches should not bring a big difference in the final similarity score. Besides the loss, the mixer decoder and reconstructing in the flow space have also already been verified to be helpful for object-centric learning.

**Questions:**

Other questions about details:
- (L38) How does the model "groups patches with similar motion into the same slot"? Neither the temporal loss nor the reconstruction loss introduces such a grouping signal.
- (L112) how are the slots $S_0$ initialized? Are they learnable, or randomly sampled from Gaussian?
- Why the performance of VideoSAUR and several baselines have better performances on MOVi-E than MOVi-C as MOVi-E should be more challenging with both moving and static objects, and ego motion? Moreover, some previous works (e.g. [1,2]) have reported worse results of SAVI++ (unconditional version) on MOVi-E, given that SAVI should have even worse numbers compared with SAVI++, I am curious why the reported numbers of SAVI are higher? Especially for the comparison with [1] given that VideoSAUR is developed based on [1]. Is it because of the number of slots?
- Conceptually, what's the role of slot mixing under the SlotMixer? It is like an additional layer of the cross-attention union (Allocation Transformer)

Ref:
[1] Bridging the Gap to Real-World Object-Centric Learning: https://openreview.net/forum?id=b9tUk-f_aG.
[2] Object discovery from motion-guided tokens: https://arxiv.org/abs/2303.15555


### Justification for rating
Though the proposed method has some limitations, I do think the proposed VideoSAUR is interesting. Currently, I hold a borderline acceptance. If the authors could fully address my first concern in the "Weakness" section, I am willing to upgrade my rating. Will also refer to other reviewers' comments.

---
The rebuttal has addressed my concern so that I upgrade my rating.

---

> ### Author Rebuttal · Authors · 2023-08-09
>
> Dear reviewer, thank you for your detailed review and interesting suggestions to improve our paper. Below we address your questions. We hope that our additional experiments will lead you to consider upgrading your rating.
>
> > One of my concerns is the use of the pre-trained ViT encoder. […] It is still interesting to see if the proposed loss is useful *without* prior knowledge. It would be useful to add an experiment that uses a self-supervised ViT *which is only pre-trained on the target dataset, e.g. MOVi-E*.
> >
>
> Thanks for this suggestion. Indeed, a strength of our method is that it does not require any additional inputs other than images, so it is natural to test how much the ImageNet data bias is needed. To test this, we pre-train a ViT-B/16 encoder with the Masked Auto-Encoder (MAE) approach for 200 epochs on the MOVi-E dataset using standard hyperparameters, and then use it for training VideoSAUR (referred to as MAE+MOVi-E) with temporal features similarity loss.
>
> The results are presented in **Table R4**. Interestingly, we observe that VideoSAUR is able to use such features for discovering objects in MOVi-E, reaching 70 FG-ARI, while not using any external data. We also use *MAE+MOVi-E* features to train VideoSAUR on MOVi-C where it reaches similar results (59.8 FG-ARI) as VideoSAUR trained with standard *MAE+ImageNet* features. Thus, even without any external data, VideoSAUR still outperforms the SAVi and STEVE baselines. Furthermore, we expect that tuning MAE hyperparameters could further improve the results.
>
> > Novelty is the other concern. Though the authors claim the temporal feature matching loss is novel, the overall idea is adapted from contrastive random walk [41].
> >
>
> We indeed were inspired by a contrastive random walk (CRW), and we do use a similar construction of the affinities, as the reviewer correctly points out. But both the loss function and the overall learning setup in CRW are different from our work. While the supervised loss $L_{sup}$ in CRW (Eq. 3) does look similar to our loss $L^{sim}$ (Eq. 5), there are fundamental differences: in $L_{sup}$, *the affinity matrix is optimized*, while in our approach $L^{sim}$, the model is trained *to predict a fixed affinity matrix*. Thus, in CRW’s $L_{sup}$, the loss is used to maximize feature similarities between matching nodes, whereas in our approach, we train a slot representation by predicting pre-constructed self-supervised feature similarities, and use the structure in these similarities to induce an object grouping. We think that this is sufficiently different from CRW (both conceptually and mathematically) to form a novel contribution.
>
> > Besides the loss, […] reconstructing in the flow space have also already been verified to be helpful for object-centric learning.
>
> While optical flow prediction has indeed been shown to be beneficial for object-centric learning, we demonstrate the novel insight that temporal feature similarities can offer similar benefits but require no flow annotations, while working sensibly for static objects and being more robust to camera motion. The novelty over optical flow prediction thus lies in broadening the kind of datasets the method can be applied to.
>
> > How does the model "groups patches with similar motion into the same slot"?
>
> Slot attention-like models essentially learn groupings that lead to efficient reconstructions under the restricted capacity of the latent slot bottleneck. When reconstructing RGB images, these groupings e.g. capture areas of similar color, because the information needed to reconstruct such an area is efficient to represent in a single slot. Analogously, something similar happens under our temporal similarity loss: when groups of patches have similar motion patterns (temporal feature similarities), it is most efficient to represent them in the same slot in order to predict the similarities well. The same principle holds for optical flow prediction as well. Does this sufficiently address your question?
>
> > How are the slots initialized? Are they learnable, or randomly sampled from Gaussian?
>
> For all VideoSAUR experiments, we use i.i.d. sampling from a Gaussian with learned mean and variance as slot initialization (see L113). This way learned object representation is invariant to the object order and also more flexible as the number of objects can be varied during inference.
>
> > Why the performance of VideoSAUR and several baselines have better performances on MOVi-E than MOVi-C [..] why the reported numbers of SAVI are higher? Especially for the comparison with [1] given that VideoSAUR is developed based on [1]. Is it because of the number of slots?
>
> As we are using 15 slots for VideoSAUR and the other baselines on MOVi-E, their performance may differ from [1] and [2]. Secondly, [1] is evaluating a video-based model on 1-frame videos, which could be a disadvantage for SAVi as it was trained on many-frame videos. In contrast, we evaluate SAVi on the whole video, leading to better results. Similarly, for MOVi-C vs. MOVi-E, we think this also is due to the number of slots, as changing the number of slots can significantly change the results (e.g. Table 13 in [1]).
>
> > Conceptually, what's the role of slot mixing under the SlotMixer?
>
> Slot mixing creates a feature vector for each spatial position (patch) that is used to reconstruct that spatial position. This is done by taking a convex combination of the slots, using different weightings for different spatial positions. The weights are computed by a dot-product between the slots and the outputs of the allocation transformer. This operation is equivalent to a single-head attention step that uses the “slots” as “values”, so indeed this could be seen as just an additional (but special) layer of the allocation transformer. We give more details on this in Appendix C.1.

---

> > ### Comment · Reviewer_L1X2 · 2023-08-10
> > **post-rebuttal comments**
> >
> > Thanks to the authors for their response. The response is well-written and can address most of my concerns.
> >
> > However, after seeing the comments from other reviewers and taking another pass at the paper, I raised two other questions:
> >
> > - let's first assume $y_t^{rec}$ is fully optimized, then in this case as long as we have perfect $y_t^{rec}$ and $y_{t+1}^{rec}$, we can compute a perfect $y_t^{sim}$. Then I am curious: if the temporal similarity loss works as a role to facilitate the learning of the reconstruction loss?
> > - as mentioned by Reviewer XQ9f, and I agree that, it is unclear about the real mechanism of the temporal consistency loss: is the performance improvement from predicting the similarity score, or from predicting **both $y_t^{rec}$ and $y_{t+1}^{rec}$** as $y_t^{sim}$ can be derived from $y_{t}^{rec}$ and $y_{t+1}^{rec}$. I think this point does matters as it affects the global positioning/picture of the paper (temporal similarity v.s. predict the future). To make it clear, slightly different from XQ9f, I think we can validate if predicting $y_{t}^{rec}$ + $y_{t+1}^{rec}$ instead of $y_{t}^{rec}$  + $y_t^{sim}$ can achieve a similar or a lower performance.
> >
> > If the authors can provide the ablation, it will be greatly helpful to better understand the proposed method!

---

> > > ### Author Response · Authors · 2023-08-16
> > > **Temporal similarity v.s. predict the future ablation**
> > >
> > > Dear reviewer, thank you for your suggestion. We agree that comparing between predicting the next frame features $y_{t+1}^{rec}$ directly and only indirectly using them for temporal features similarity $y_{t}^{sim}$ is indeed valuable to disentangle the source of improvement behind our method.
> > >
> > > The additional experimental results are presented in Tables R5-R7.  We find that using our proposed temporal feature similarity ($y_t^{rec} + y_t^{sim}$) **brings clear additional performance benefits** (e.g., +13 FG-ARI MOVi-C; +6 FG-ARI MOVi-E) in comparison with next frame feature prediction ($y_t^{rec} + y_{t+1}^{rec}$). In addition, VideoSAUR with the combination of current and next frame feature reconstruction ($y_t^{rec} + y_{t+1}^{rec}$) performs significantly better than just current frame feature reconstruction ($y_t^{rec}$) (+7 FG-ARI on MOVi-C; +2 FG-ARI on YouTube-VIS), showing that predicting the future is generally helpful. We conclude that both the future prediction task and the specific way it is implemented in our temporal feature similarity are important to achieve the final VideoSAUR performance on all the datasets.
> > >
> > > As for *why* the temporal feature similarity is better than pure next frame feature prediction, we believe it is because the similarity loss requires predicting relationships between features, which is not needed for pure feature prediction. In addition, the similarity prediction task does potentially contain more signal to optimize, as it removes unnecessary details (noise) from the prediction of particular next frame features. Thus, even though $y_t^{sim}$ could be derived from a perfect prediction of $y_t^{rec}$ and $y_{t+1}^{rec}$, in practice, optimizing it directly focuses the model on different aspects of the targets that turn out to be useful for object discovery.
> > >
> > > **Implementation details for next frame feature prediction.** Reconstructing frame features $y_t^{rec}$ and $y_{t+1}^{rec}$ simultaneously with a single decoder is problematic, because the decoder masks we use for evaluation would be in reference to both the current and next frame. One way to overcome this problem is by using two decoders $d_{current}$ and $d_{future}$, each producing their own predictions and masks. In this case, masks from $d_{current}$ can be used for evaluation. While more powerful, this approach also requires more memory and is slower. In our experiments, we confirm that the version with two different Mixer decoders performs better than simultaneous reconstruction with one decoder (see rows 1 and 2 in Tables R5-R7). We use this better version for our comparisons even though it is heavier than our method that needs only one decoder.
> > >
> > > > if the temporal similarity loss works as a role to facilitate the learning of the reconstruction loss?
> > > >
> > >
> > > Regarding this question, note that the temporal similarity loss can also be used as a standalone loss, and we show that optimizing $y_t^{sim}$ on its own yields similar results to optimizing both $y_t^{sim}$ and $y_t^{rec}$ on MOVi-C in Table 2. We also checked if optimizing $y_t^{sim}+y_t^{rec}$ leads to a lower reconstruction error compared to optimizing just $y_t^{rec}$ and found this not to be the case. Thus we conclude that the value of the temporal similarity is not just in helping the model to reconstruct better, but actually requires the model to learn additional information.
> > >
> > >
> > > ### Table R5. Future frame features reconstruction ablation on MOVi-C
> > > | Method                                                     | Video FG-ARI | Video mBO |
> > > |------------------------------------------------------------|--------------|-----------|
> > > | Feat. Rec. + Next Frame Feat.Rec                           | 44.6 |	23.5      |
> > > | Feat. Rec. + Next Frame Feat.Rec (two decoder heads)       |47.2 |	24.7     |
> > > | Feat. Rec. + Temp. Sim.                                    | **60.8** |	**30.5**    |
> > >
> > > ### Table R6. Future frame features reconstruction ablation on MOVi-E
> > > | Method                                                     | Video FG-ARI | Video mBO |
> > > |------------------------------------------------------------|--------------|-----------|
> > > | Feat. Rec. + Next Frame Feat.Rec                           | 61.3         | 22.1      |
> > > | Feat. Rec. + Next Frame Feat.Rec (two decoder heads)       | 62.9         | 24.0      |
> > > | Feat. Rec. + Temp. Sim.                                    | **69.2**         | **26.0**      |
> > >
> > > ### Table R7. Future frame features reconstruction ablation on YouTube-VIS
> > > | Method                                                     | Video FG-ARI | Video mBO |
> > > |------------------------------------------------------------|--------------|-----------|
> > > | Feat. Rec. + Next Frame Feat.Rec                           | 33.4	| 24.6    |
> > > | Feat. Rec. + Next Frame Feat.Rec (two decoder heads)       | 37.9  |	27.3      |
> > > | Feat. Rec. + Temp. Sim.                                    | **39.5**         | **29.1**      |

---

> > > > ### Comment · Reviewer_L1X2 · 2023-08-20
> > > >
> > > > Thanks for the updated results! They are promising and I now don't have other concerns.

---

### Official Review · Reviewer_tzwP · 2023-07-08

**Soundness:** 3 good
**Presentation:** 4 excellent
**Contribution:** 3 good
**Rating:** 8
**Confidence:** 3

**Summary:**

This paper proposes VideoSAUR for unsupervised video object segmentation / grouping. The key idea proposed in this paper is to use a temporal feature similarity loss, in combination with a feature reconstruction loss. The grouping is implemented with recurrent Slot Attention. Experiments are conducted on various benchmarks (MOVi-C, MOVi-D, MOVi-E, YT-VIS-2019, YT-VIS-2021, and DAVIS) with good improvement over existing methods. Written presentation is clear and easy to follow.

**Strengths:**

* The paper has strong experimental results: (i) solid improvements over existing methods; and (ii) a comprehensive set of ablations with different insights about the problem as well as the proposed method.

* Written presentation is clear, consistent and easy to follow and understand.

* The idea of using temporal feature similarity is not new (Vondrick et al ECCV'18, Vondrick et al CVPR'16, may be some others), but this paper shows an effective way to use it with self-supervised features, and recurrent Slot-Attention, yield strong results in unsupervised video-object segmentation.

**Weaknesses:**

* I don't find any issue with this paper, may be further evaluate on more challenging dataset such as UVO will make this work srtonger.

* There may be a stress test to push the method into an extreme setting, e.g., a very long video of hundreds of frames, to see when the recurrent Slot Attention will be failed. This may provide hints for future work.

**Questions:**

* As mentioned above, I would be interested in seeing (1) how this proposed approach works on more challenging datasets; (2) how this method deals with extremely long videos (when will it fail); (3) any discussion on how to deal with long videos would be nice.

**Limitations:**

* The reviewer does not foresee any potential negative societal impact of this work.

---

> ### Author Rebuttal · Authors · 2023-08-09
>
> We thank the reviewer for their very positive feedback! We are glad you like the paper for its “strong experimental results” and that you find the paper “clear, consistent and easy to follow”. We answer your questions below.
>
> > How does the proposed approach works on more challenging datasets?
> >
>
> We agree that taking VideoSAUR to even more complex datasets is very interesting! Similarly, it would also be intriguing to see how the method scales with increasing dataset size. However, given that we already are the *first* unsupervised object-centric method to use the Youtube-VIS dataset, we think that this is out of scope for this paper.
>
> > Dealing with long videos? When will it fail? How to deal with long video?
> >
>
> We thank the reviewer for the suggestion. Indeed better performance on long videos is one of the potential future directions for our method.
>
> As we already discussed in Appendix B.1 and the Conclusions section there are several limitations of our current approach on long videos. First, the number of slots is static and needs to be chosen a priori, whereas, in the long videos, the objects frequently approach and disappear from the scene. Next, slots can get reassigned to each frame and can bind to different objects or the background. We visualize those failures on a long video (see in **Figure R1**).
>
> To overcome these limitations, several innovations are needed. First, it is important to discover and maintain objects’ slots that are currently not visible in the image. This could be done with memory mechanisms and maintaining if each slot is active or not as an additional external variable (similar to $z_{present}$ in AIR-based [1] object-centric methods). Next, a slot re-identification module that matches any new objects to previously discovered, but not visible ones could further improve the performance on long videos. Finally, learning a generative latent object-centric world model (with set-based latent dynamics) could further improve the consistency of slots for object-centric representations on long videos.
>
> [1] Attend, Infer, Repeat: Fast Scene Understanding with Generative Models

---

> > ### Comment · Reviewer_tzwP · 2023-08-15
> > **Thank you for the rebuttal**
> >
> > Thanks for the discussions. Please include the new figure R1 and discussion (if the space allows) if the paper gets accepted. I keep my rating unchanged.

---

### Official Review · Reviewer_avhJ · 2023-07-08

**Soundness:** 3 good
**Presentation:** 4 excellent
**Contribution:** 3 good
**Rating:** 7
**Confidence:** 3

**Summary:**

The paper considers the problem of unsupervised video-based object-centric learning. It incorporates a temporal feature similarity loss that encodes temporal correlations and introduces a motion bias for object discovery. This loss helps to achieve state-of-the-art performance on the synthetic MOVi dataset. The model is able to learn video object-centric representation on the YouTube-VIS dataset in a fully unsupervised way.

**Strengths:**

1) The paper is very well-written and it is pleasant to read. The visualizations and graphics are particularly nice and done with good care.
2) The proposed VideoSAUR method significantly improves performance on synthetic video datasets over related works of SAVi and STEVE. It also to learn video object-centric representation on unconstrained real-world videos of YT-VIS.
3) The ablation study is extensively done. It considers all the main parts of the method along with the most important hyperparameters.

**Weaknesses:**

I enjoyed the paper and I think that the paper is already in good shape for acceptance.

**Questions:**

No additional questions.

**Limitations:**

The limitations are briefly discussed at the end of the paper.

---

> ### Author Rebuttal · Authors · 2023-08-09
>
> We thank the reviewer for their positive feedback! We are glad you find the paper to be “very well-written” and “pleasant to read”. We hope that the reviewer would find additional experiments requested from other reviewers interesting. Their description could be found in the general response.

---

### Author Rebuttal · Authors · 2023-08-09

We thank the reviewers for their feedback and appreciate that they found our paper "well-written", "pleasant to read", and "well-organized". In addition, the reviewers recognized that our "comprehensive ablation study” is “extensively done” and that it brings insights into the method’s performance. In addition, the proposed approach was recognized as “sound” and “effective” and, as a result, yielded strong outcomes.

In addition, we also value the reviewer’s constructive feedback, with requests for additional experiments and ablations. Below, we list the main new experiments that we present in this rebuttal:

- We compare our method with the state-of-the-art image-based DINOSAUR method and several baselines on the YT-VIS dataset showing that VideoSAUR performs better than DINOSAUR on this dataset (**Table R1 in the Rebuttal’s PDF**).
- We provide an extended ablation study on the MOVi-E dataset, where we ablate both the decoder choice and the proposed loss, showing that both are needed for the state-of-the-art performance of our method. In addition, we showed that VideoSAUR significantly outperforms the extension of DINOSAUR to the video domain (**Table R2 in the Rebuttal’s PDF**).
- We study the performance of our method when GT optical flow is used as a target. We show that the motion information of each pixel alone is a worse grouping signal than the proposed temporal similarity of *highly semantic* self-supervised features (**Table R3 in the Rebuttal’s PDF**).
- We investigate our method's performance when no additional datasets are available for self-supervised pertaining. We showed that using MAE pre-trained only on MOVI-E allows VideoSAUR to achieve similar performance to the MAE pre-trained on ImageNet on both MOVi-E and MOVi-C datasets, showing that object-centric bias in ImageNet dataset is not necessary for successful object discovery with VideoSAUR (**Table R4 in the Rebuttal’s PDF**).
- Finally, we visualize and discuss how our method can fail on long videos and discuss potential future work needed to enable our method to work on long videos (**Figure R1 in the Rebuttal’s PDF**).

We are happy that the reviewers recognize the following contributions:

- This paper shows an effective way to use it **[temporal feature similarity] with self-supervised features** and recurrent Slot-Attention (tzwP)
- It incorporates a temporal feature similarity loss that encodes temporal correlations and introduces a motion bias for object discovery (avhj)
- The proposed VideoSAUR model yields state-of-the-art performance on several datasets (L1X2)

In addition, we would like to add:

- **Efficient video architecture integration**: We integrated this loss with the efficient SlotMixer decoder, where this loss synergistically reduces optimization difficulties.

________________________________________________
For convinience we also include the same tables as in PDF below:

**Table R1. Image-based comparison on YouTube-VIS**

| Method     | Image FG-ARI    |
|------------|-----------------|
| SAVi       | 13.2 ± 5.0      |
| STEVE      | 25.3 ± 1.8      |
| DINOSAUR   | 39.2 ± 0.3      |
| VideoSAUR  | 43.1 ± 0.4      |


**Table R2. Extended ablation of VideoSAUR components on MOVi-E**
| Decoder | Loss        | FG-ARI | mBO  |
|---------|-------------|--------|------|
| Mixer   | Feature Reconstruction  | 62.3   | 20.6 |
| MLP     | Feature Reconstruction  | 68.6   | 27.6 |
| MLP     | Feature Similarity  | 74.5   | 28.8 |
| Mixer   | Feature Similarity  | 74.1   | 34.1 |

**Table R3. Ablation of VideoSAUR features similarity targets with GT optical flow**

| VideoSAUR                     | MOVi-C | MOVi-E |
|-------------------------------|--------|--------|
| w/ GT Optical Flow (backward) | 48.1   | 28.9   |
| w/ GT Optical Flow (forward)  | 48.9   | 30.1   |
| w/ Feature Similarity         | 60.7   | 73.9   |

**Table R4. Self-supervised method (Masked Auto-Encoder) pretraining on MOVi-E**


| VideoSAUR           | FG-ARI on MOVi-C| mBO on MOVi-C | FG-ARI on MOVi-E | mBO on MOVi-E |
|---------------------|---------------|------------|---------------|------------|
| w/ MAE+ImageNet   | 58.0          | 30.4       | 72.8          | 27.1       |
| w/ MAE+MOVi-E     | 59.8          | 27.5       | 70.6          | 23.3       |

---

### Decision · Program_Chairs · 2023-09-21

**Decision:**

Accept (poster)

**Comment:**

Reviewers have come to a consensus on accepting this paper,  the authors are expected to incorporate the suggestions from reviewers in the final camera ready version.